# Genome-wide profiling in colorectal cancer identifies PHF19 and TBC1D16 as oncogenic super enhancers

Qing-Lan Li[1,3], Xiang Lin[1,3], Ya-Li Yu[2,3], Lin Chen[1,3], Qi-Xin Hu[1], Meng Chen[2], Nan Cao[2], Chen Zhao[1], Chen-Yu Wang[1], Cheng-Wei Huang[1], Lian-Yun Li[1], Mei Ye [2✉] & Min Wu [1✉]

Colorectal cancer is one of the most common cancers in the world. Although genomic mutations and single nucleotide polymorphisms have been extensively studied, the epigenomic status in colorectal cancer patient tissues remains elusive. Here, together with genomic and transcriptomic analysis, we use ChIP-Seq to profile active enhancers at the genome wide level in colorectal cancer paired patient tissues (tumor and adjacent tissues from the same patients). In total, we sequence 73 pairs of colorectal cancer tissues and generate 147 H3K27ac ChIP-Seq, 144 RNA-Seq, 147 whole genome sequencing and 86 H3K4me3 ChIP-Seq samples. Our analysis identifies 5590 gain and 1100 lost variant enhancer loci in colorectal cancer, and 334 gain and 121 lost variant super enhancer loci. Multiple key transcription factors in colorectal cancer are predicted with motif analysis and core regulatory circuitry analysis. Further experiments verify the function of the super enhancers governing *PHF19* and *TBC1D16* in regulating colorectal cancer tumorigenesis, and KLF3 is identified as an oncogenic transcription factor in colorectal cancer. Taken together, our work provides an important epigenomic resource and functional factors for epigenetic studies in colorectal cancer.

[1] Frontier Science Center for Immunology and Metabolism, RNA Institute, Hubei Key Laboratory of Cell Homeostasis, Hubei Key Laboratory of Developmentally Originated Disease, Hubei Key Laboratory of Intestinal and Colorectal Diseases, College of Life Sciences, Wuhan University, Wuhan, Hubei 430072, China. [2] Division of Gastroenterology, Department of Geriatrics, Hubei Clinical Centre & Key Laboratory of Intestinal and Colorectal Diseases, Zhongnan Hospital, Wuhan University, Wuhan, Hubei 430072, China. [3] These authors contributed equally: Qing-Lan Li, Xiang Lin, Ya-Li Yu, Lin Chen. ✉email: wumeiye08@163.com; wumin@whu.edu.cn

The binding of transcription factors (TFs) to enhancers is one of the critical steps in transcription activation. Recently, the development of epigenomics revealed novel features of active and silent enhancers and shed light on the study of transcriptional regulation in multiple research fields[1–5]. Epigenetic marks on chromatin are important signatures for cell identification, which co-operate with transcription factors to regulate transcription[2,6,7]. Histone modifications mark enhancers on chromatin and are critical for their activity. H3K4me1 is the mark for primed enhancers;[7,8] H3K27ac for active enhancers and H3K27me3 for poised enhancers[1]. Though the initial discovery was concluded from ChIP-Seq of mediator subunits, now H3K27ac in the intergenic chromatin is widely used for the identification of active enhancers[6,9,10]. Moreover, it was discovered that many genes are often regulated by multiple enhancers and the state of these enhancers varies in different cell types[1,4]. Therefore, it has emerged as critical questions for many fields how enhancer activity is regulated for signaling pathways and selective gene transcription.

Pioneer studies hypothesized that gain of enhancer activity is one of the common features for cancers[6,11–13], which is supported by some recent studies in patients and animal models[10,14,15]. However, it is still not clear whether it is a common feature for all cancers or just a portion of them. Interestingly, many genes related with epigenetic regulation of enhancer activity are frequently mutated in cancer, such as *lysine methyltransferase 2 C/D* (*KMT2C/D*, also named as *MLL3/4*), *E1A binding protein p300* (*EP300*), *CREB binding protein* (*CEBBP*), *lysine demethylase 6 A* (*KDM6A*, also named as *UTX*) and *lysine demethylase 5 C* (*KDM5C*)[12,16–20]. Moreover, inhibitors for bromodomain-containing 4 (BRD4), one reader of H3K27ac on enhancer, were shown to be effective in cancer treatment[21]. It is then urgent to clarify the roles of enhancers in cancer and the underlying mechanisms.

It has been shown that the enhancers controlling the transcription of key oncogenic genes, such as *MYC proto-oncogene* (*MYC*), distinguish in different types of cancers[22]. It is probably that transcription factors activated by signaling networks vary a lot in different cancer cells, which causes that enhancers are activated in different ways in response to variant genome mutations and upstream signals. Thus, enhancer profiling may reflect the feature of distinguished cancers and be used for classification[10,13,23–25]. Genome-wide profiling of active enhancers has been carried out in several types of cancers, such as pancreatic cancer, nasopharyngeal carcinoma, and clear cell renal carcinoma, usually using the approach of H3K27ac ChIP-Seq[10,14,26]. A pan-cancer study using TCGA RNA-Seq data also analyzed global enhancer distribution[24], but since a large portion of enhancer RNA is difficult to detect, the approach is not very reliable.

Colorectal cancer (CRC) is one of the most common cancers in the world. Recent studies about aberrant DNA methylation have gained sight of the diagnosis community, and epigenetic regulation becomes one of the critical regulatory factors for CRC[27–32]. Some groups have studied the genome-wide distribution of active enhancers in CRC[23,33]. The early studies used H3K4me1 as a mark which was not suitable to identify functional active enhancers[23]. H3K4me1 marks primed enhancers, which does not represent functional enhancers in the genome. A recent study took normal colonic epithelial crypts, CRC cell lines, and four primary patient colorectal tumors, and analyzed the genome-wide difference of active enhancers using H3K27ac ChIP-Seq[33]. Flebbe et al. also performed H3K27ac in a small number of rectal cancer tissues, but did not analyzed cancer-specific enhancer features[34]. These studies used very few clinical samples, which could not reflect the real clinical features of CRC tissues and are not very helpful to enhancer studies in CRC.

In this work, to establish a comprehensive map for active enhancers in CRC, we perform H3K27ac ChIP-seq analysis with 73 pairs of CRC tissues (tumor tissues with paired adjacent native tissues), as well as the corresponding genomic and transcriptomic sequencing. We identify thousands of enhancers and multiple TFs involved in CRC, and a portion of them are experimentally verified, which provides important epigenomic resources and research candidates for future studies in CRC.

## Results

**Genome-wide study of enhancer distribution in CRC patient tissues.** To establish a comprehensive genome-wide view of active enhancers of CRC patient tissues, we totally collected 80 pairs of tissues (tumor and their adjacent tissues) from CRC patients. The tissues were collected from patients who received surgical treatment at Zhongnan Hospital of Wuhan University (Wuhan, China), and no specific criteria were applied when collecting tissues. The patients were mostly from the Huazhong area of China, especially Hubei province. We optimized the ChIP-Seq protocol and performed H3K27ac ChIP-Seq for these samples, as well as the corresponding mRNA and input DNA sequencing. Some samples failed in the study, and eventually, we got high quality of sequencing data from 74 CRC tissues and 73 native tissues, among which 73 were paired (Fig. 1A, Supplementary Fig. S1 & Supplementary Data 1). We also performed H3K4me3 ChIP-Seq for 43 pairs of tissues. Totally we generated 524 high-quality sequencing samples, including 147 H3K27ac, 86 H3K4me3, 144 RNA-Seq, and 147 genomic sequencing samples (Sup. Data 2), which provide important epigenomic information for CRC studies.

For bioinformatic analysis, we first identified active enhancers in each tissue by H3K27ac peaks far away from transcription start sites (TSS). To avoid interference by artificial results, the enhancers at least appearing in two samples were considered as significant enhancers. Our analysis revealed totally of 27,156 significant enhancers in native tissues and 39,207 in tumor tissues, most of which were distributed on introns and intergenic regions as expected (Fig. 1B & Supplementary Fig. S2A). Meanwhile, we identified 9896 and 10663 active promoters in native and tumor tissues, respectively (Supplementary Fig. S2A). The saturation analysis showed that the gained enhancers in tumors reached 80% when using less than 40 pairs of samples for analysis, and 90% with around 50 pairs, indicating the sample size used in our study was good enough for statistical analysis (Fig. 1C). We downloaded H3K4me1 (ENCODE, ENCFF557VIT) and BRD4 (GEO, GSM3593876) ChIP-seq data in HCT116, a CRC cell line, from public databases, and calculated the RPM values of H3K4me1 and BRD4 signal in significant CRC tumor enhancer loci. Our analysis showed that the H3K27ac peaks of our study are nicely correlated with the published BRD4 and H3K4me1 signal (Sup. Fig. S2B&C). Our RNA-Seq analysis identified 2226 up-regulated differently expressed genes (DEGs) and 1979 down-regulated DEGs in CRC tumors (Fig. 1D). Compared with the TCGA data, many DEGs are overlapped (Sup. Fig. S2D). The proportion of genes assigned with multiple enhancers was shown (Sup. Fig. S2E). *MYC* is a well-known oncogene[22], and we found that in 63 pairs of patient tumor tissues (87.5%), MYC expression is more than 2 times higher than that in the corresponding native tissues. H3K27ac track on its enhancer was shown as an example (Fig. 1E). In the adjacent native tissues, *MYC* expression was very low, and the H3K27ac signal on its enhancer and eRNA were close to the background; in tumor tissues, *MYC* was highly expressed, accompanied by elevation of H3K27ac on its enhancer and eRNA expression (Fig. 1E). One early study has reported active enhancers in multiple CRC cell lines and a few CRC samples[33].

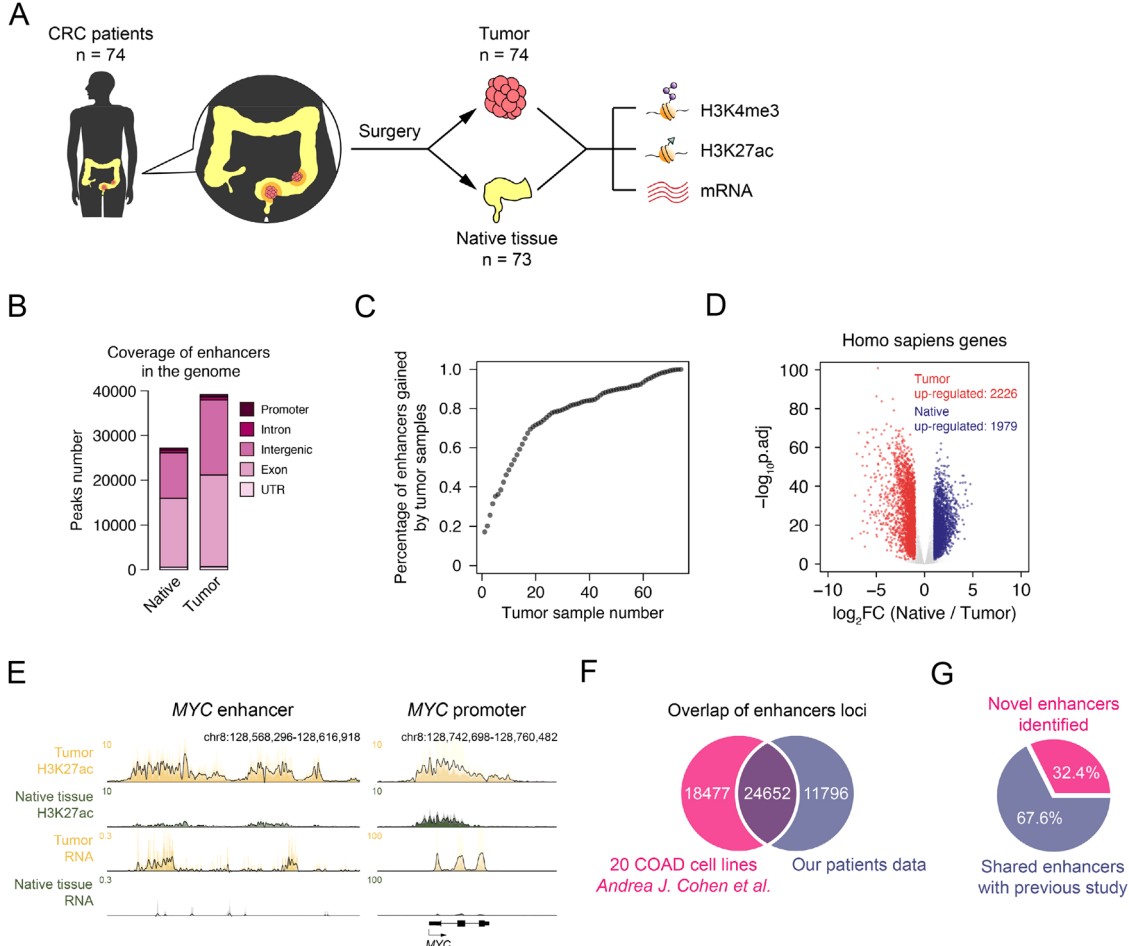

**Fig. 1 The annotation of active enhancers in CRC patient tissues. A** Experimental workflow for studying the enhancer landscapes of tumor and native tissues from CRC patients. **B** Genomic distribution of enhancer elements in tumor and native tissues from CRC patients. **C** Saturation analysis showing the percentage of newly gained enhancers comparing with total significant enhancers along with an increasing number of the tumor samples. **D** Fold change (FC) and p.adj of human gene expression comparing tumor and native tissues. Red dots represent tumor up-regulated genes, blue dots for native tissue up-regulated genes, and grey dots for genes not changed. **E** Normalized ChIP-seq and RNA-seq Meta tracks showing H3K27ac and mRNA signal on *MYC* promoter and enhancer loci. **F** Overlap of enhancer loci between our patient data and 20 COAD cell lines (GSE77737, Andrea J. Cohen et al.). **G** Percentage of novel enhancers in CRC identified in our study.

Comparison of the two studies revealed that we identified 11796 different active enhancers in CRC, which was 32.4% of the total enhancers (Fig. 1F, G).

**Identification of variant enhancer loci in tumor.** To identify significant active enhancers specific in tumors, we first compared the enhancers of all samples and found that some tissues had a relatively low number of H3K27ac peaks (less than 2,500), or variant enhancer loci (VELs, less than 500) compared with the corresponding adjacent tissues (Supplementary Fig. S3A and B). We considered it was probably due to the sampling problem. Since when we were using random pieces of clinical dissected tissues for experiments, it was possible that some tissue samples only contained very few cells, or a large part of their cells actually were very similar as their corresponding ones. So, we ruled them out in the following statistical analysis. We totally identified 6690 significant VELs, including 5590 gain VELs and 1100 lost VELs (Fig. 2A, Supplementary Data 3–5) and the pipeline was shown in Supplementary Fig. S3C. At the recurrence threshold of 14 and 19 patients, 95% of the gain and lost VELs achieved statistical significance ($q$-value<0.1, paired $t$-test, with Benjamini-Hochberg correction; Fig. 2B). Supporting the reliability of these analysis, significant gain VELs exhibited higher H3K27ac level in

tumors than native tissues, and opposite in significant lost VELs (Supplementary Fig. S3D and E). Meanwhile, genes associated with gain VELs showed elevated expression in tumors in comparison with native tissues, while lost VEL-associated genes were broadly repressed in tumors; and the magnitude of the change in expression positively correlated with the number of VELs per gene (Supplementary Fig. S3F). The gain VELs close to *IL20RA* and *FOXQ1* and the lost VELs close to *PPARGC1B* were shown as representative (Fig. 2C, Supplementary Fig. S3G and H). The identified VELs could nicely distinguish the native and tumor tissues (Fig. 2A). Human disease ontology and GO analysis showed that the associated genes of gain VELs were highly related with CRC (Fig. 2D & Supplementary Fig. S3I), while those of lost VELs were related with normal colon functions (Supplementary Fig. S3J). To further evaluate the potential of H3K27ac or enhancer information in distinguishing tumor and normal tissues, we clustered the adjacent native and tumor tissues with PCA using the information of gene expression, H3K27ac on enhancers, and H3K4me3 on promoters. The adjacent and tumor samples were nicely distinguished using both gene expression or significant enhancers, while the different H3K4me3 peaks did not (Fig. 2E–G), suggesting enhancer information is useful for tumor identification.

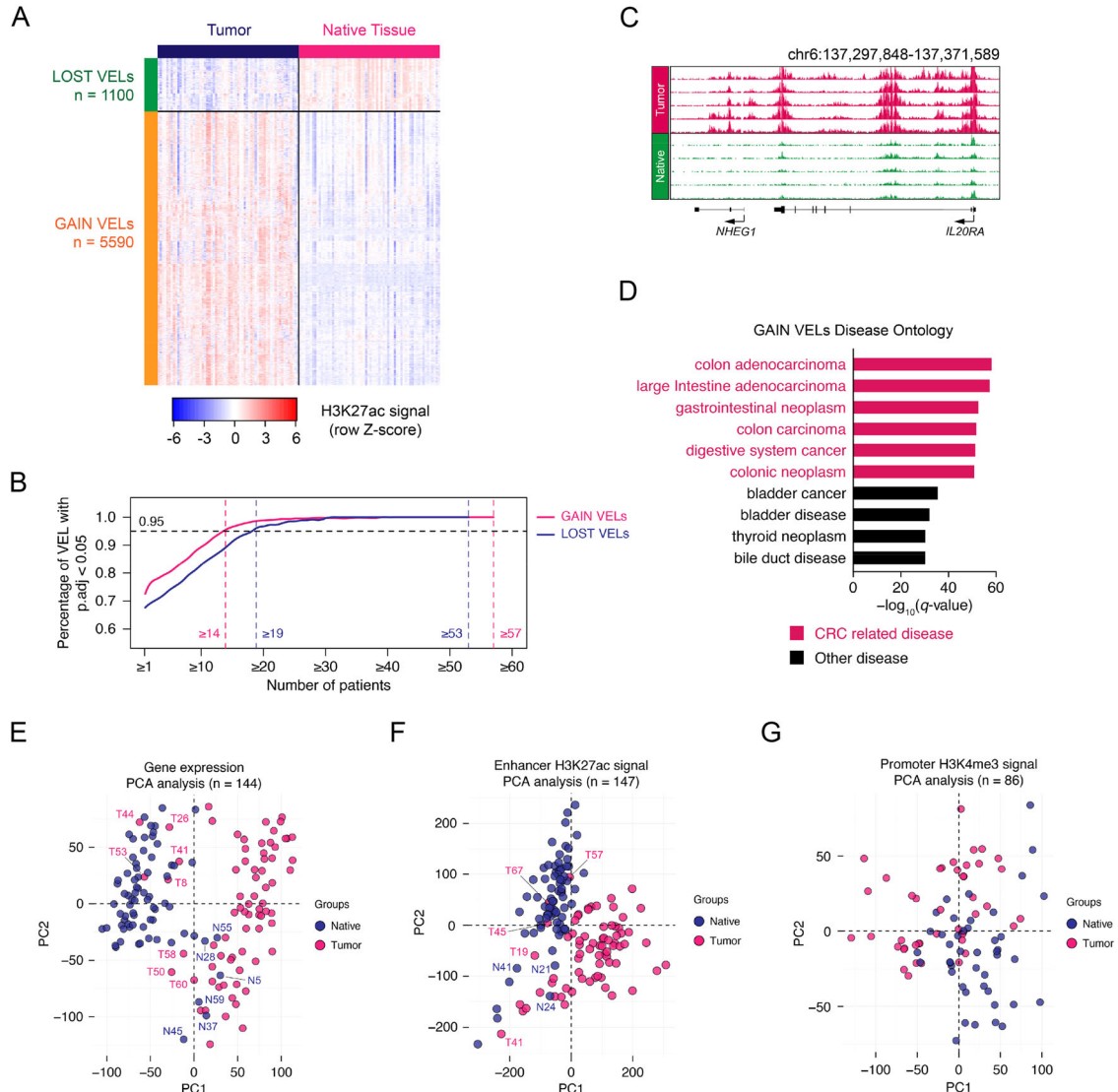

**Fig. 2 Identification of variant enhancer loci in CRC. A** Relative H3K27ac signals of lost and gain VELs in all tumor and native tissues. **B** The required recurrence for gain and lost VELs meeting statistical significance (p.adj <0.05). The two vertical dashed lines at left highlights the recurrence of gain and lost VELs when achieve the cut-off (0.95, black dashed line) of significant percentage, and the two lines at right highlights the highest recurrence in tumor or native tissue of gain and lost VELs, respectively. For gain VELs, when recurrence reach to 14, the percentage of significant VELs is 95.635%; for lost VELs, when recurrence reaches to 19, the percentage of significant VELs is 96.273. p.adj indicates the BH adjusted *t*-test *p*-value. A two-sided test was utilized here. **C** Representative H3K27ac tracks of gain VEL on *IL20RA* loci. **D** The human disease ontology in which gain VELs participated detected by GREAT software (version 3.0.0). The red bars represent CRC-related diseases and the black bars represent other diseases. **E–G** PCA analyses to classify tumor and native tissues using gene expression (**E**), all significant enhancers (**F**), and promoters (**G**) information identified using our patient RNA-seq and ChIP-seq data.

**The enhancer features of CRC subgroups**. To further investigate the enhancer features in CRC subgroups, we utilized one of the common approaches, the consensus molecular subtypes (CMS) classification of CRC tumors[35], and classified patients into four subgroups (Fig. 3A and B, Supplementary Fig. S4B–C). The correlation analysis based on the identified VELs and subgroup-specific VELs showed that the tissues of the CMS2 group had the highest correlation, while CMS4 was the lowest (Fig. 3C and D, Supplementary Fig. S5), suggesting CMS4 might be more heterogeneous than others. Interestingly, when comparing the enhancers among the four subgroups, we found that CMS2 had the largest number of active enhancers, significant gain VELs, and specific gain VELs (Fig. 3E–G, Supplementary Fig. S6A). The average H3K27ac signal of gain VELs in CMS2 was also higher than the other three (Fig. 3F). While, the four groups had no big

difference at the amount of gain VELs among individual samples, as well as the correlation of H3K27ac across the genome (Supplementary Fig. S6B–D). The above study indicates that the CMS2 group is more homogenous than others; and it has more specific active enhancers, which might be a novel feature for it.

Then, we studied the function of VEL-associated genes, and identified enhancers and genes specifically activated in each subgroup. Some representative enhancers and genes for each group were shown (Supplementary Fig. S6E–L). For CMS2, we found its specific gain-VEL-associated genes are mainly involved in WNT signaling, cell migration, and lipid metabolic process (Fig. 3H, Supplementary Fig. S6M). Activation of WNT signaling and enhanced cell migration are expected, since *APC* is one of the most frequent mutated genes in CRC and cell migration is a hallmark for cancer cells[36,37]. Lipid metabolism was linked with

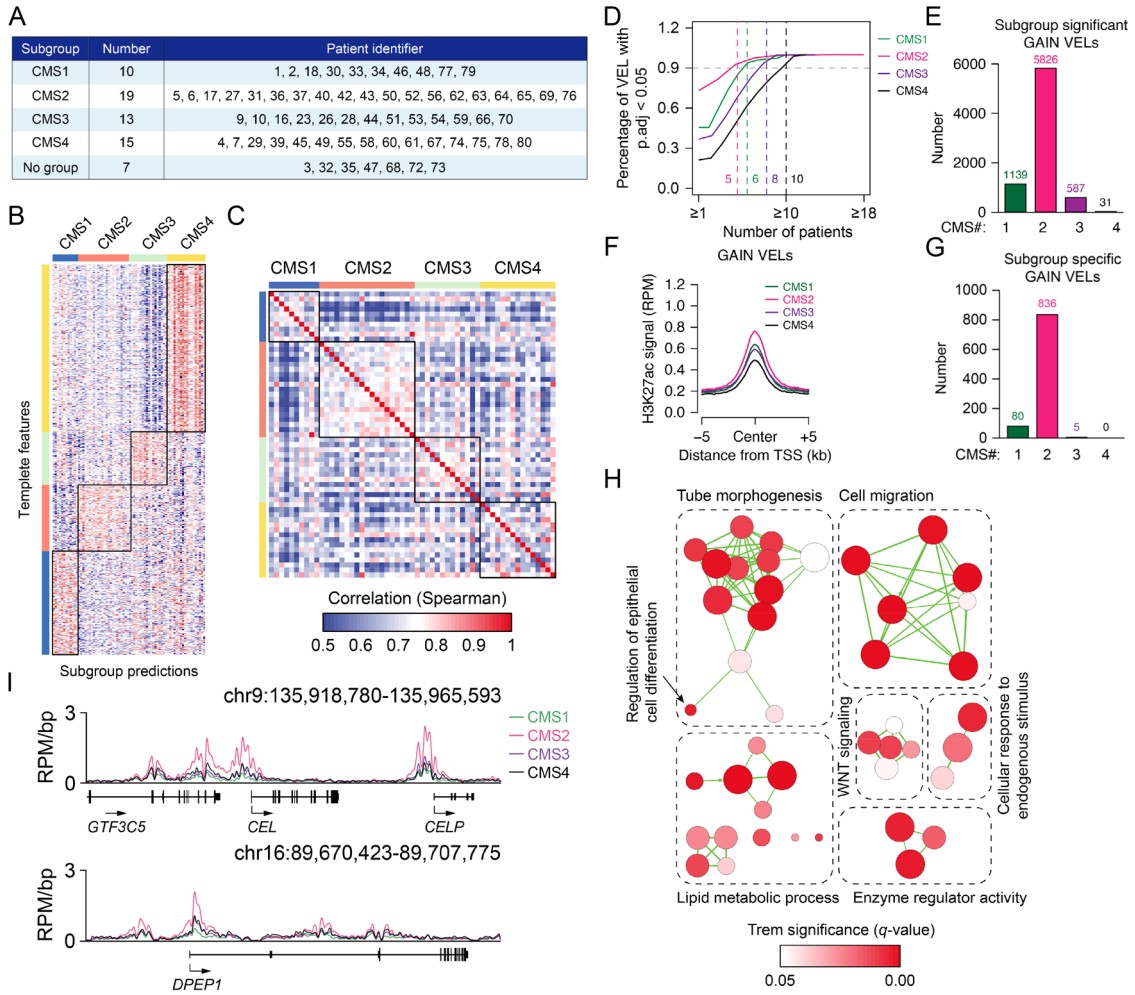

**Fig. 3 The feature of enhancers in CMS subgroups. A** The patient identifier of members in four CMS subgroups. **B** The consensus molecular subtypes (CMS) classification of CRC samples using R package CMScaller. **C** Correlation of H3K27ac signal on the regions of gain VELs in all tumor samples of CMS1–4 subgroups. Correlations were calculated by the Spearman correlation coefficient. **D** The required recurrence for gain VELs in each CMS subgroup to meet statistical significance (p.adj <0.05) at different cut-offs. The dashed lines highlight the recurrence of gain VELs when achieve the cut-off (0.9, black dashed line) of a significant percentage. When recurrence reach to the cut-off, the significant percentage of CMS1 = 93.942%, CMS2 = 93.615%, CMS3 = 95.741% and CMS4 = 93.548%. p.adj indicates the BH adjusted *t*-test *p*-value. A two-sided test was utilized here. **E** The number of subgroup significant gain VELs in four CMS subgroups. **F** The average H3K27ac signal (RPM) on the regions of gain VELs in four CMS subgroups. **G** The number of subgroup-specific gain VELs in each CMS. The subgroup-specific gain VELs were identified when the mean RPM of one VEL in one CMS subgroup was 1.5 times higher than the other three. **H** Functional annotation of target genes associated with CMS2 specific gain VELs based on their significant overlap with gene sets annotated in Gene Ontology (Biological Process) and pathway database (Reactome). **I** Meta tracks of normalized H3K27ac on *CEL* and *DPEP1* gene loci in four CMS subgroups.

CRC but ambiguous results from different groups exist[38–40]. Our analysis suggested dysregulation of lipid metabolic homeostasis is possibly associated with certain CRC subgroups. Some VELs of the CMS2 subgroup and their associated genes were shown, including those involved in lipid metabolism, such as *CEL* and *DPEP1* (Fig. 3I, Supplementary Fig. S7).

**Analysis and verification of variant super enhancer loci.** Activation of oncogene-associated super-enhancers is one of the important features for cancer[6]. Using similar approaches as VEL identification, we identified the variant super-enhancers loci (VSEL) in tumor tissues, including 334 gain VSELs and 121 lost VSELs, among which several well-known oncogenic targets were identified, such as *MYC*, *VEGFA*, and *LIF* (Fig. 4A, Supplementary Fig. S8A and B, Supplementary Data 6&7). H3K27ac level on the gain VSELs were significantly increased and decreased on the

lost VSELs as expected (Fig. 4B and C). As expected, genes associated with gain VSELs expressed higher than those associated with lost VSELs (Supplementary Fig. S8C). We utilized the H3K27ac value on VSELs to cluster CRC patients, together with normal intestinal tissues (Supplementary Fig. S8D and E). The analysis distinguished the native and tumor tissues, and classified CRC patients into three subgroups (Supplementary Fig. S8D and E). The results suggest VSELs might be useful for CRC classification. We compared the two classification approaches, and found that most of the CMS2 group samples were classified into G1 subgroup based on VSELs (Supplementary Fig. S8F), supporting our previous conclusion that the CMS2 group is more homogenous than others.

To experimentally verify the functions of the identified VSELs, we compared the H3K27ac profiles on top gain VSELs of CRC tissues with those in HCT116 cells. The gain VSELs appearing in HCT116 were chosen and the dCas9-KRAB system was utilized

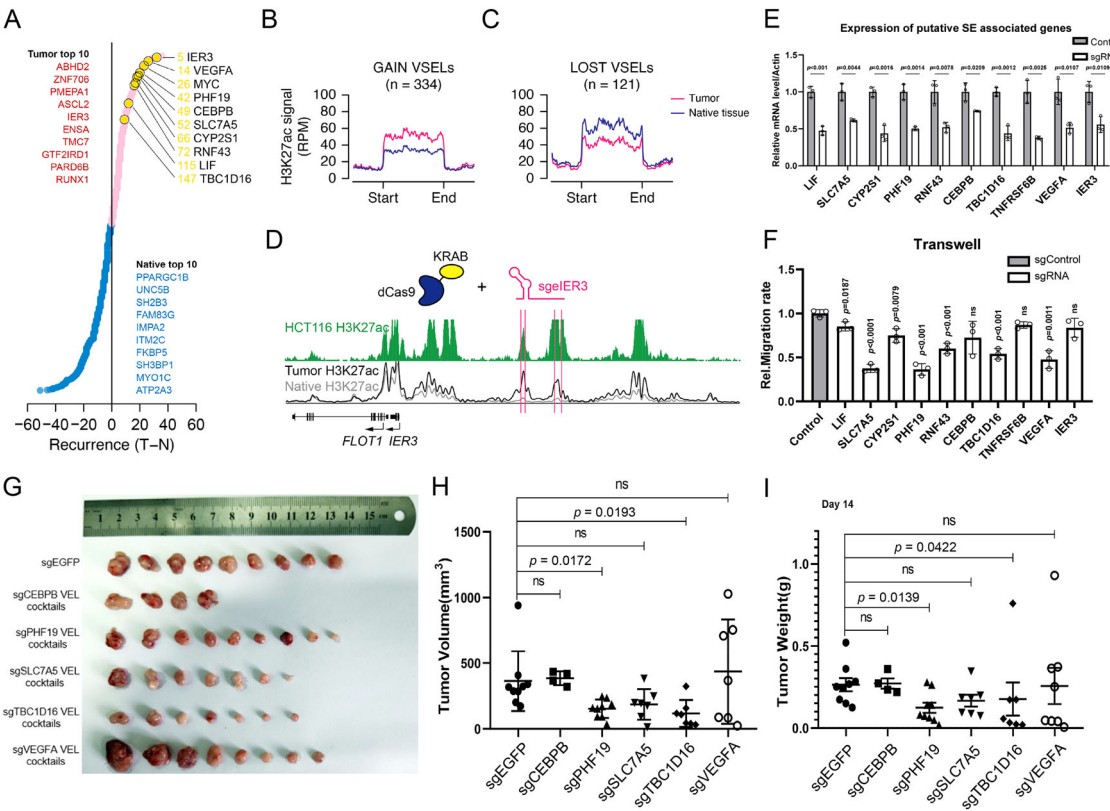

**Fig. 4 Functions of tumor-specific super enhancers in CRC. A** The genes associated with top super-enhancers (SEs) ranked by recurrence. Red dots represent tumor-specific SE genes and blue dots represent native tissue-specific SE genes. Top 10 tumor and native tissue-specific genes were listed. **B**–**C** The average H3K27ac signal (RPM) at the regions of gain VSELs (**B**) and lost VSELs (**C**) in tumor and native tissues. **D** Meta normalized H3K27ac tracks at IER3 gene loci. The green track on the top represents the H3K27ac signal in HCT116, and the black and grey lines at the bottom represent the average signal of tumor and native tissues, respectively. The pink lines indicate the target positions of dCas9-KRAB sgRNAs. **E** Bar plot showing the relative mRNA level of *LIF, SLC7A5, CYP2S1, PHF19, RNF43, CEBPB, TBC1D16, TNFRSF6B, VEGFA,* and *IER3* in control and sgRNA groups ($n = 3$). A sgRNA control targeting EGFP was used as a control in the following experiments. $*p < 0.05$. **F** Transwell assays for HCT116 cell lines stably transfected with dCas9-KRAB sgRNAs of the enhancers mentioned in Fig. 5E ($n = 3$). **G**–**I** Xenograft experiments in nude mice were performed with HCT116 stable cells expressing the indicated sgRNAs. The tumors were pictured (**G**), and their volume and weight were shown (H&I). $n = 9$ for all groups. Data are presented as mean values ± SEM. Statistical analysis was performed using a two-sided Student $t$ test. $p$ value was labelled on the corresponding items.

to repress enhancer activity[41]. Since one SE usually covers a relatively large chromatin region, we design multiple sgRNAs for each SE and made stable cell lines in HCT116, RKO or SW620, three CRC cell lines (Fig. 4D, Supplementary Fig. S9, Supplementary Data 8). Totally we analyzed 11 SEs, among which 10 SEs were found to regulate the expression of their proximal genes in multiple cell lines, including *IER3, LIF, SLC7A5, CYP2S1, PHF19, RNF43, CEBPB, TBC1D16, TNFRSF6B,* and *VEGFA* (Fig. 4A and E, Supplementary Fig. S9). For some enhancers, the expression of multiple close genes was repressed (Sup. Fig. S9). We also measured the H3K27ac level on some loci and found that dCas9-KRAB/sgRNA effectively repressed H3K27ac on the enhancers of *CEBPE, CYP2S1, IER3, PHF19, RNF43,* and *TBC1D16* (Supplementary Fig. S10A). In comparison with their expression in CRC patient tissues Supplementary Fig. S10B), our data indicated that the above SEs regulate the expression of the corresponding genes. Moreover, we established stable cell lines of repressed enhancers in HCT116 and studied their proliferation and migration ability. The difference of proliferation and cell cycle was not very significant (Supplementary Fig. S11A and B), however, quite a few cell lines exhibited attenuated migration ability in multiple cell lines, including *PHF19, LIF, SLC7A5, CYP2S1, RNF43, VEGFA,* and *TBC1D16* (Fig. 4F, Supplementary Fig. S11C–F). To further investigate the functions of the above SEs in CRC, we performed xenograft assays with stable HCT116

cells, which showed that repression of PHF19 and TBC1D16 SEs significantly reduced tumor growth, and the expression of target genes was confirmed in randomly selected tumors (Fig. 4G–I, Supplementary Fig. S11G–K). When SLC7A5 SE was repressed, tumor growth also showed a trend of reduction, but the difference was not significant enough (Fig. 4G–I).

**Predication and verification of functional transcription factors.** To investigate the potential transcription factors (TFs) playing key roles in CRC, the DNA sequences of VELs was used for prediction by HOMER software. The top hits of gain and lost VELs were listed (Fig. 5A and Supplementary Fig. S12A and B). The hypothesis of core regulatory circuitry was raised to identify core TFs in cells[42,43]. To improve TF prediction, we utilized the method to identify key TFs in CRC tissues (Fig. 5B&C, Supplementary Fig. S12C). ASCL2 was predicted as a CRC-specific TF with the highest score (Fig. 5C). The H3K27ac level of *ASCL2* enhancer greatly increased in tumors (Supplementary Fig. S12D), and the gene expression analysis based on TCGA datasets suggested *ASCL2* was highly expressed specific in colorectal cancer (Supplementary Fig. S12E). These suggest ASCL2 is a key TF in CRC, which is consistent with the previous publications[44–46]. Since ASCL2 has been characterized in CRC, we did not further investigate its roles.

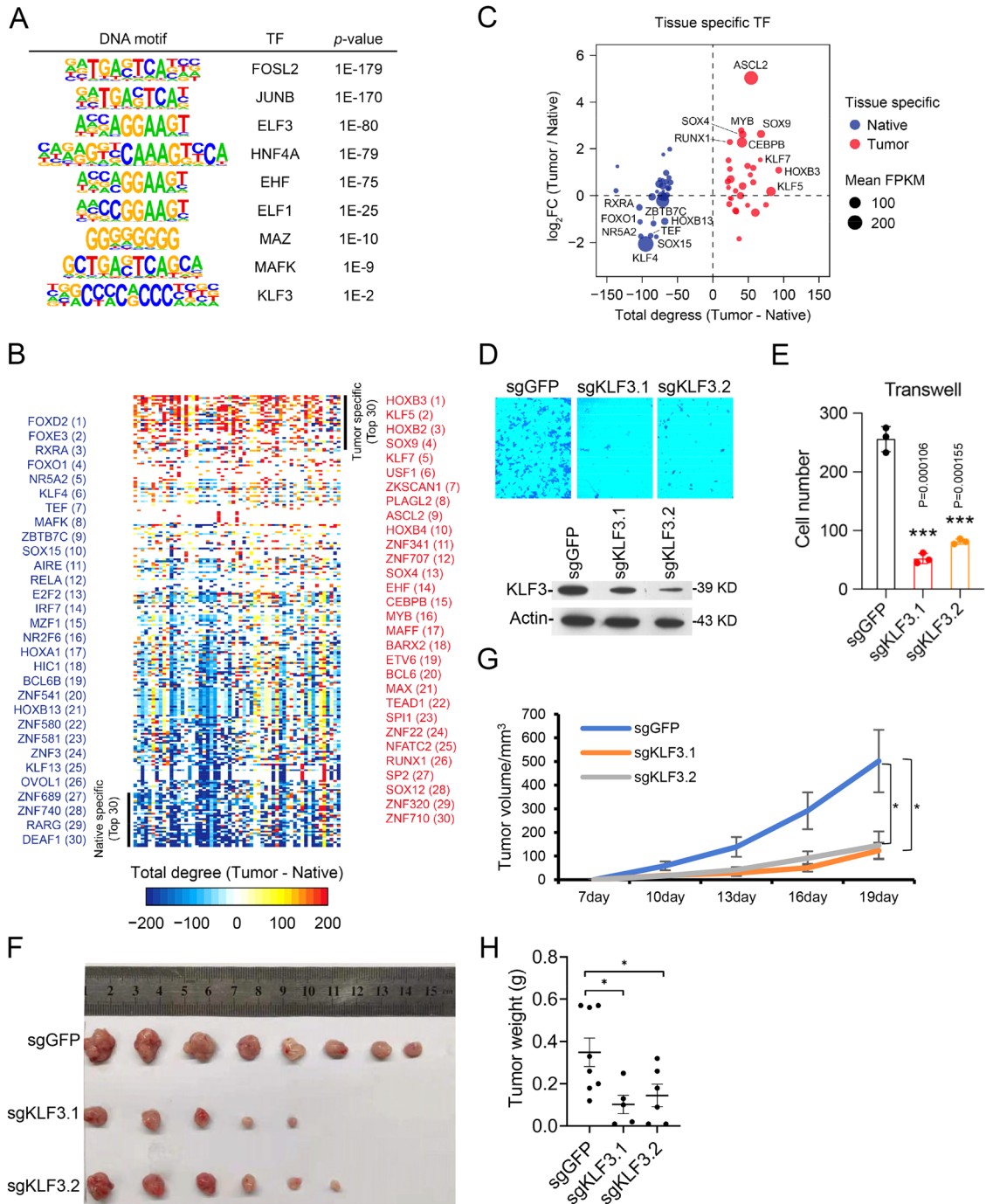

**Fig. 5 Prediction of functional transcription factors in CRC. A** DNA motifs enriched within nucleosome-free regions (NFRs) of tumor gain VELs determined by HOMER motif analysis. **B** Heatmap of transcription factors ranked by predicted core regulatory circuitry (CRC) total degrees (Tumor - Native tissue). Top 30 tumor and native-specific TFs were listed. **C** Scatter plot showing the total degree (Tumor-Native tissue) and expression FC (Tumor/Native tissue) of the specific TFs listed in Fig. 5B. Blue dots represent top 30 tumor specific TFs, and red dots represent the top 30 native-specific TFs. Circle size indicates the mean expression (FPKM) of TFs in its specific tissue. **D, E** Transwell assay for HCT116 cell line with *KLF3* knockdown. KLF3 was measured with western blotting. *p*-value = 1.06E−4 (sgKLF3.1), and 1.55E−4 (sgKLF3.2). *n* = 3. Data are presented as mean values ± SEM. **F–H** *KLF3* stably knockdown HCT116 cells by sgRNA were injected into nude mice (10^6 cell pear mouse, *n* = 10). Tumors were pictured (**F**), and tumor growth curve (**G**) and weight (**H**) were shown as mean (± SEM). *P*-value in G for sgKLF3.1 and KLF3.2 are 0.0261 and 0.0296, in **H** for sgKLF3.1 and KLF3.2 are 0.0256 and 0.0355, respectively. Statistical analysis was performed using a two-sided Student *t* test. *$p < 0.05$, **$p < 0.01$.

Combining the above results and the published literatures, we selected around 10 TFs for experimental verification. Unfortunately, some genes express low in the selected cell lines or knockdown experiments did not work for some genes. Eventually, we only got 4 genes successfully knocked down, including *KLF3*

and *MAFK*, two novel TFs, *MAZ* and *RUNX1*, two recently reported TFs functioning in CRC but not well characterized[46–49]. The significance of the selected TFs in each patient was calculated (Supplementary Fig. S12F). Knockdown of these genes did not affect cell proliferation in cell proliferation assay; while *KLF3*,

*MAZ,* and *RUNX1* knockdown by siRNAs, or *KLF3* knockdown by CRISPR/sgRNAs, repressed cell migration, but not *MAFK* (Fig. 5D and E, Supplementary Fig. S13, S14A and B). Our results identified KLF3 as a TF involved in CRC, and confirmed the reported roles of RUNX1 and MAZ. MAFK was found at the side of native tissues in core regulatory circuitry analysis (Fig. 5B), and our experimental results did not find its role in proliferation or migration, suggesting *MAFK* is not involved in CRC although it was predicted in DNA motif assay. To further confirm the function of *KLF3,* we knocked it down with CRISPR/sgRNAs in HCT15 and RKO cell lines, and the similar results were observed (Supplementary Fig S14C–F). Xenograft assay was then performed with *KLF3*-deficient HCT116, and the result indicated that *KLF3* knockdown repressed tumor growth in mice (Fig. 5F–H).

## Discussion

CRC is one of the most common cancers in the world. Though early screening greatly improves the curative ratio, novel classification approaches and drugs are still urged to be developed. The current study provides a comprehensive map of H3K27ac and active enhancers in CRC patients. The early studies used cell lines to determine CRC-specific enhancers[33]. Our study used paired patient tissues, which provided much more reliable data and identified many VELs and VSELs. Moreover, we experimentally confirmed the roles of more than 10 SEs in CRC. These provide important information for CRC research.

Our analysis predicted many TFs functioning in CRC, such as KLF3. ASCL2 was reported to be an oncogene in CRC, and our results showed that ASCL2 is highly expressed and H3K27ac on it increased in CRC tissues. Our results were consistent with published studies. We also experimentally confirmed the roles RUNX1 and MAZ, and identified KLF3 as an oncogenic TF in CRC. Further studies about these TFs will provide important information for CRC research. Loss of *KLF3* was previously shown to be associated with more aggressive CRC[50]. We then analyzed the expression of *KLF3* in our data and TCGA data. We did not observe a significant difference between the adjacent and cancer tissues in our collection; among TCGA samples, KLF3 expression was higher in tumor tissues but not significant enough. We then compared the expression between the adjacent and cancer tissues of four CMS groups. Interestingly, KLF3 is lower expressed in the cancer tissues of the CMS1 group, and higher expressed in the CMS3 group (Supplementary Fig. S14I). Although our sample number is a little bit small when split into four groups, the results suggest that KLF3 may play different roles in the four groups. It will be interesting to determine KLF3s function in each CRC subgroup.

Interestingly, verification of the identified VSELs and TFs indicated that most of them were more related with cell migration, but less with proliferation. It is possible that the normal intestinal epithelial cells are growing rapidly, and the change of epigenomic marks on migration-related genes is much bigger than proliferation genes, which caused the top candidates are mostly related to migration. Other possibilities could be that the CRC samples we collected are mostly at relative late stages (Supplementary Fig. S1A & Supplementary Data 1), and the tumor cells were probably at the metastasis stage or ready for it. The difference of enhancers and genes governing migration was probably much more significant than other genes between paired tissues, and the chosen VSELs were all among the most significant ones.

Interestingly, our study showed that SE repression sometimes caused down-regulation of multiple proximal genes, and the affected genes might vary in different cell lines (Supplementary

Fig. S9F, G and I). It showed the complexity of transcription regulation by SEs, and indicated one fact that one SE is consisted of multiple TF binding sites and could be bound by multiple TFs, which may govern the transcription of multiple associated genes.

It was believed that the gain of H3K27ac on the oncogenic enhancers is a common feature for cancers[21]. Our analysis indicated that only the CMS2 group has the obvious feature of enhancer activity elevation, and the other subgroups have much less gained VELs. These indicate that in CRC, the global increase of active enhancers is an important feature for just one subgroup, not for all.

The homeostasis of lipid metabolism has been linked with CRC for many years, however, the detailed mechanisms is not clear and the use of statin analogues failed in CRC treatment[38]. We found that H3K27ac significantly increased on the enhancers of genes related with lipid metabolism. The bioinformatics analysis also pointed out that the gain VELs of the CMS2 subgroup were enriched with genes involved in lipid metabolic processes. So, it is possible that the dysregulation of lipid homeostasis is only associated with the CMS2 group, which should be explored by the future studies.

## Methods

**Ethics approval and consent to participate**. A total of 80 pairs of primary tumor tissues and corresponding adjacent tissues were collected from patients who received surgical treatment at Zhongnan Hospital of Wuhan University (Wuhan, China) between August 2017 and February 2018. Written informed consent for the usage of samples in the current study was obtained from the patients before surgery. Samples of the collected tissues were preserved in liquid nitrogen. Clinical case data of patients was also collected. The collection procedures of clinical specimens were approved by the Clinical Research Institution Review Committee and Ethics Review Committee of Zhongnan Hospital, and consent of each patient was obtained before collection.

**Animal housing and ethics approval**. BALB/C-nu/nu mice were purchased from Gempharmatech, and 5-weeks old male mice were used in the studies. All the animal operations were following the laboratory animal guidelines of Wuhan University and were approved by the Animal Experimentations Ethics Committee of Wuhan University (Protocol NO. 14110B). All the mice were born and maintained under the pathogen-free condition at 20–24 °C with a humidity of 40–70% and a 12/12 h dark/light cycle (lights on at 7:00 AM, lights off at 7:00 PM), with free access of water and food (Animal Center of College of Life Sciences, Wuhan University).

**Reagents and cell lines**. Antibodies recognizing H3K4me3 (Millipore (Merck), 04–745), H3K27ac (Abcam, ab4729, RRID: AB_2118291), and KLF3 (Abclonal A7195, RRID: AB_2767745), were purchased from indicated commercial sources. Protein G-Sepharose beads were from GE Healthcare. PCR primers were custom synthesized by BGI and siRNAs by GenePharma. HCT116, RKO, and HCT15 cell lines were purchased from Cell Bank of Chinese Academy and cultured under recommended conditions according to the manufacturer's instruction with 10% FBS.

**ChIP assay and ChIP-sequencing**. ChIP assay was performed as previously described[51]. Briefly, around sixty milligrams of each tissue were cut into 1 mm$^3$ pieces in PBS with protease inhibitor. Tissue pieces were cross-linked for 10 min at room temperature with 1% formaldehyde and then quenched with 0.125 M of glycine for 5 min. Cross-linked tissues were triturated for 30 s and then centrifuged at 10,000 g, 4 °C for 5 min. Supernatant with massive oil was discarded and the precipitates were lysed with 1 mL lysis buffer (50 mM Tris-HCl pH 8.0, 0.1% SDS, 5 mM EDTA) for 4 min with gentle rotation. After centrifugation at 10,000 g, 4 °C for 2 min, the pellet was washed once with digestion buffer (50 mM Tris-HCl pH 8.0, 1 mM CaCl2, 0.2% Triton X-100), incubated in 630 µL digestion buffer with 1 µL MNase (NEB, M0247S) at 37 °C for 20 min and quenched with 8 µL 0.5 M EDTA. The resulted mixture was sonicated and the pellet was discarded after centrifugation. 30 µL supernatant was taken for checking the efficiency of digestion. Immunoprecipitation was performed with 150 µL sheared chromatin, 2 µg antibody, 50 µL Protein G beads and 800 µL dilution buffer (20 mM Tris-HCl pH 8.0, 150 mM NaCl, 2 mM EDTA, 1% Triton X-100, 0.1% SDS) overnight at 4 °C. Next day, the beads were washed once with Wash buffer I (20 mM Tris-HCl pH 8.0, 150 mM NaCl, 2 mM EDTA, 1% Triton X-100, 0.1% SDS), once with Wash buffer II (20 mM Tris-HCl pH 8.0, 500 mM NaCl, 2 mM EDTA, 1% Triton X-100, 0.1% SDS), once with Wash buffer III (10 mM Tris-HCl pH 8.0, 250 mM LiCl, 1 mM EDTA, 1% Na-deoxycholate, 1% NP-40) and twice with TE (10 mM Tris-HCl pH

8.0, 1 mM EDTA). The beads were eluted twice with 100 μL elution buffer (1% SDS, 0.1 M NaHCO3, 20 mg/mL Proteinase K) at room temperature. The elution was incubated at 65 °C for 6 h and then purified with DNA purification kit (TIANGEN DP214-03). Primers for ChIP-qPCR were listed in Sup. Data 8.

ChIP-seq libraries were constructed with ChIP and input DNA using VATHS Universal DNA Library Prep Kit for Illumina (Vazyme ND606). Briefly, 50 μL of DNA (8–10 ng) was end-repaired for dA tailing, followed by adaptor ligation. Each adaptor was marked with a barcode of 8 bp DNA. Adaptor-ligated DNA was purified by AMPure XP beads (1:1) and then amplified by PCR of 9 cycles with the primer matching with adaptor universal part. Amplified DNA was purified again using AMPure XP beads (1:1) in 35 μL EB elution buffer. For multiplexing, libraries with different barcode were mixed with equal molar quantities (30–50 million reads per library). Libraries were sequenced by Illumina Nova-seq platform with pair-end reads of 150 bp.

The DNA sequencing information of ChIP-seq Input samples was considered as the corresponding Whole Genome Sequencing (WGS) data. Before the immunoprecipitation procedure in ChIP assay, about 10% of the sonicated lysate supernatants were taken as the input samples, which contained fragmented genomic DNA (around 150 bp length) and binding proteins. Input sample was added with elution buffer (1% SDS, 0.1 M NaHCO3, 20 mg/mL Proteinase K) to 100 μL, incubated at 65 °C for 6 h, and then purified with DNA purification kit (TIANGEN DP214-03). The aim of this procedure is to remove DNA-binding proteins and get pure DNA. VATHS Universal DNA Library Prep Kit for Illumina (Vazyme ND606) was used to prepare libraries for genomic DNA, and the libraries were sequenced by the Illumina Nova-seq platform with pair-end reads of 150 bp.

**RNA-sequencing**. Around 40 mg tissue was used for RNA extraction using the Ultrapure RNA Kit (CWBIO, CW0581M). Briefly, tissues were triturated for 30 s in 1 mL TRIzon provided in the kit, incubated at room temperature for 5 min, added with 200 μL chloroform, and shaken drastically. After centrifugation at 10,000 g, 4 °C for 10 min, the upper water phase was moved into an adsorption column provided by the kit. The column was then eluted with 50 μL RNase-free water. RNA-seq libraries were constructed by NEBNext Poly(A) mRNA Magnetic Iso-lation Module (NEB E7490) and NEBNext Ultra II Non-Directional RNA Second Strand Synthesis Module (NEB E6111). Briefly, mRNA was purified with poly-T magnetic beads and first and second-strand cDNA was synthesized. The resulted cDNA was purified by AMPure XP beads (1:1) and eluted in 50 μL nucleotide-free water. The subsequent procedures were the same as described in ChIP-seq library construction, except that the sequencing depth was 20 million reads per library. RNA-seq libraries were sequenced by the Illumina Nova-seq platform with pair-end reads of 150 bp.

**ChIP-seq data processing**. The adaptor sequence was removed using Cutadapt (version 1.16) to clean ChIP-seq raw data. Cleaned reads were mapped into the human reference genome (hg19) using BWA (version 0.7.15) with default settings. Peak calling for tissues was performed by MACS2 with a p-value threshold of 1E-10. The patients with a peak number of less than 2,500 were excluded from further analysis, no matter in native or tumor tissue (Patient 20, 21, 22, and 24 were excluded).

We calculated the normalized RPM as the ChIP-seq signal in a specific region. Briefly, ChIP-seq reads aligning to the region were extended by 200 bp and the density of reads per bp was calculated using Python package HTSeq (version 0.9.1). The density of reads in each region was normalized to the total number of million mapped reads, producing read density in units of reads per million mapped reads per bp (RPM per bp).

**Plotting meta representation of ChIP-seq signal**. Considering the sample number of our patient data, we utilized a way of calculating the mean to compactly display the integrated H3K27ac ChIP-seq signal in specific groups. For an individual region, we calculated the aligned read number per bp within this region using the R package HTSeq mentioned above, and then normalized to RPM. H3K27ac ChIP-seq signal is smoothed using a simple spline function and plotted as a translucent shape or a line in units of RPM per bp.

**RNA-seq data processing and DEG identification**. The adaptor sequence was removed using Cutadapt (version 1.16) to clean RNA-seq raw data. Cleaned reads were aligned to the human reference genome (hg19) using HISAT2 (2.1.0) with default settings. Uniquely aligned reads were counted at gene regions using the package featureCounts (version 1.4.6) based on Gencode v19 annotations. Differ-ential gene expression analysis between native and tumor tissue was performed using the R/Bioconductor package DESeq2 (version 1.26.0) with contrast adjust-ment for multiple groups comparison. We considered the pairs information of all samples when using DESeq2, and used a design of the form "~ patient + condi-tion" to account for the pairs when providing the Sample Table. The "patient" column was used to record the patient identifier and the "condition" column to indicate the tissue type. Genes whose log2FC < 1 and $p_{adj}$ < 1E−2 were identified as differential expressed genes (DEGs).

**Promoter, enhancer and super enhancer analysis**. For both H3K4me3 and H3K27ac ChIP-seq data, peaks that could not be identified in at least two same kind of tissues were excluded from further analysis. H3K4me3 peaks located within the region surrounding ± 2.5 kb of transcriptional start sites (TSS) were identified as promoters; and H3K27ac peaks away from the ± 2.5 kb flank region of TSS were identified as enhancers. The promoters and enhancers of each samples were merged into one single set. Super enhancers were identified as following: firstly, super-enhancers (ROSE) algorithm was used to classify and rank sets of two or more H3K27ac peaks (detected by MACS2, p-value < 1E-10) within 12.5 kb dis-tance and further than 2.5 kb from a transcriptional start site; secondly, a plot was graphed and a tangent line of the curve was drawn with the slope value of 1; finally, the enhancers above the point of tangency were defined as super-enhancers. HOMER module annotatePeaks.pl was used to calculate the number of enhancers located in different chromatin elements.

**Identification of VELs**. To identify the significant VELs between native and tumor tissue, we first identified all VELs in paired native and tumor tissues. Individual sample VEL were defined as enhancers whose H3K27ac fold change (FC) was larger than >2 between native and tumor tissues. The patients with VEL numbers (GAIN + LOST VELs) less than 500 were excluded from further analysis. We merged all VELs into one single coordinate file, and calculated the recurrence and significance (Benjamini–Hochberg corrected p-value) for all VELs. We used recurrence of 14 and 19 as significance threshold for gain and lost VELs, respec-tively, because gain and lost VELs achieved the significant percentage cut-off (0.95) when recurrence larger than these numbers.

**Identification of VSELs**. For variant super-enhancer loci (VSEL), the identifying procedure was similar as described above in "Identification of VELs". If the VSELs number in an individual patients was less than 10, the patient would be excluded from further analysis (Patient 52 and 67 were excluded). And the significant percentage cut-off was changed to 0.9.

**Identification of genes associated with VSELs**. SE-associated Genes were identified by rose2 (https://github.com/linlabbcm/rose2) software and all these genes were merged into a single list. We considered the variation of a SE-associated gene by calculating the variant recurrence generated by its recurrence in tumor minus which in native tissue.

**Principal component analysis**. We performed PCA for gene expression, enhancer H3K27ac, and promoter H3K4me3 in native and tumor tissues. For gene expres-sion, we quantified sequencing fragments as reads per kilobase per million (FPKM) in each sample. And for two ChIP-seq signals, we used RPM. R package Facto-MineR (version 2.3) was used to perform PCA analysis.

**Human disease ontology and GO analysis**. The coordinate file of GAIN and LOST VELs were submitted to GREAT website (version 3.0.0) and the results of human disease ontology and GO analysis (biological process) were obtained for plotting.

**CMS classification for tumor tissues**. Consensus Molecular Subtype (CMS) classification of tumor tissues was performed by an R package CMScaller (version 0.99.2). With an integrated CRC tumors RNA-seq result file, this package could classify all samples into 5 subgroups (CMS1/2/3/4 and no group). The samples excluded in previous steps were not analyzed here.

**Identification of CMS subgroup specific gain VELs**. For an individual gain VEL, if the H3K27ac signal on the corresponding region in a CMS subgroup was 1.5 times higher than the other 3 subgroups, we called it a specific gain VEL for this subgroup. For all CMS subgroups, significant GAIN VELs were identified as the procedure described above in "Identification of VELs" and the significant per-centage cut-off was changed to 0.9.

**Pathway analysis for CMS2 specific GAIN VELs**. Functional characterization of CMS2 specific gain-VEL-associated genes was conducted using the ClueGO plugin for Cytoscape (version 3.8.0). These tested genes were queried against a compen-dium of gene sets from GO (Biological Process), KEGG, and REACTOME to identify significantly enriched processes and pathways. Analyses were performed using the GO Term Fusion option in ClueGO and only processes/pathways with a p-value <0.01 (right-sided hypergeometric test) following p-value correction (Bonferroni step down) were visualized.

**Prediction of enriched TFs on VELs**. HOMER software plugin findMotifsGen-ome.pl was used to calculate the significance of TFs enrichment. For VELs of all patients, the coordinate files of gain and lost VELs were used for calculation and the size parameter is 200. For VELs of the individual patient, TFs enrichment significance were calculated using nucleosome-free regions (NRFs) within VEL. NFRs were generated by PARE (version 0.08) with default settings.

**Core regulatory circuitry for super enhancer associated TFs**. To investigate the interaction network of transcription factor regulation, we calculated the inward and outward binding degree of all SE-associated TFs. All SE-associated genes annotated to encode a transcription factor were considered as the node-list for network construction. For a given TFi, the IN degree was defined as the number of TFs with a binding motif within the proximal super-enhancer or promoter of TFi. The OUT degree was defined as the number of TF-associated super-enhancers containing an enriched binding site for TFi. The IN and OUT degree were generated by crc software (https://github.com/linlabcode/CRC) and the total degree was defined as IN degree plus OUT degree.

**CRISPR-Cas9-KRAB mediated repression of VSELs**. Site-specific single guide RNAs (sgRNAs) targeting VSELs were designed with publicly available filtering tools (https://zlab.bio/guide-design-resources) to minimize off-target cleavage. For CRISPR interference, sgRNAs were cloned into the pLH-spsgRNA2 (Addgene, #64114) through the BbsI site according to the protocol recommended by Addgene. Lentivirus was generated by transfecting HEK293T cells with sgRNA expression cocktails or pHAGE dCas9-KRAB-MeCP2, together with helper plasmids, psPAX, and pMD2.G. After 12 h, cells were washed twice with PBS and a fresh medium was added. Medium-containing virus was collected 48 or 72 h after transfection, and filtered with 0.45 μm filters (Millipore). Stable cell lines were generated by infecting HCT116 with lentivirus expressing dCas9-KRAB-MeCP2 and sgRNAs. Cells were then screened with puromycin (1 μg/ml, Amresco) and hygromycin (200 μg/ml, Roche) for 48 h, and examined by western blot and RT-qPCR.

**Reverse transcription and quantitative PCR**. Cells were scraped down and collected with centrifugation. Total RNA was extracted with an RNA extraction kit (Aidlab) according to the manufacturers manual. Approximately 1 μg of total RNA was used for reverse transcription with a first-strand cDNA synthesis kit (Toyobo). The resulted cDNA was then assayed with quantitative PCR. β-actin was used for normalization. The sequences of primers are in Sup. Data 8. Assays were repeated at least three times. Data were shown as average values ± SD of at least three representative experiments. P value was calculated using the students t test.

**Cell proliferation assay**. The proliferation of colorectal cancer cells in vitro was measured using the MTT assay. Briefly, 1,000 cells were seeded into 96-well plate per well. Six well of each group were detected every day. MTT (0.25 μg) was put into each well and incubated at 37°C for 4 h. The medium with the formazan sediment was dissolved in 50% DMF and 30% SDS (pH4.7). The absorbance was measured at 570 nm. Assays were repeated at least three times. Data were shown as average values ± SD of at least three representative experiments and p value was calculated using student's t test.

**Transwell assay**. $1 \times 10^5$ HCT116 cells were plated in medium without serum or growth factors in the upper chamber with a Matrigel-coated membrane (24-well insert; pore size, 8 μm; BD Biosciences), and medium supplemented with 10% fetal bovine serum was used as a chemoattractant in the lower chamber. After 36 h of incubation, cells that did not invade through the membrane were removed by a cotton swab. Cells on the lower surface of the membrane were stained with crystal violet and counted. Assays were repeated at least three times. Data were shown as average values ± SD of at least three representative experiments and p value was calculated using students t test.

**Xenograft experiments in mice**. 5-week-old male BALB/c nude mice were purchased from GemPharmatech Co., Ltd. Colon cancer model was established by injecting subcutaneously $8 \times 10^5$ HCT116 cells per site into the flank regions of the mice. Tumor volumes were measured for two or three days once using calipers. Tumor volumes were calculated as $V = 0.5 \times length \times width^2$. Around 19 days after injection, the tumors were harvested and weighed.

**Statistics and reproducibility**. For experiments other than NGS sequencing, at least three times for each experiment were performed with similar results and different biological replicates. Data are presented as mean values ± SEM. Statistical analysis was performed using a two-sided Student t test. p value was either labelled on the corresponding items or listed in the legends. For western blotting, the original films were shown in Sup. Fig. S15.

**Reporting Summary**. Further information on research design is available in the Nature Research Reporting Summary linked to this article.

## Data availability

The raw RNA-seq and ChIP-seq data used in this study are available in the GEO database under accession number GSE156614 and raw H3K27ac and H3K4me3 ChIP-seq data are available under accession number GSE156613. The processed data are available in the supplemental tables. 20 colorectal cancer cell lines H3K27ac ChIP-seq data: GEO dataset - GSE77737. HCT116 BRD4 ChIP-seq data: GEO dataset -

GSE126221. HCT116 H3K4me1 ChIP-seq data: ENCODE – ENCFF557VIT. Public cancer patient RNA-seq data were downloaded from TCGA database (TCGA-COAD, TCGA-READ, TCGA-BLCA, TCGA-BRCA, TCGA-GBM, TCGA-HNSC, TCGA-KIRC, TCGA-LAML, TCGA-LUAD, TCGA-LUSC, TCGA-OV, and TCGA-UCEC) [https://www.cancer.gov/about-nci/organization/ccg/research/structural-genomics/tcga]. The analyzed data, including significant gain and lost VELs, significant gain and lost VSELs, were listed in Sup. Data 3–7.

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

## Acknowledgements
This work was supported by grants from Ministry of Science and Technology of China (2016YFA0502100), National Natural Science Foundation of China to L.L. (31670874), and M.W. (81972647 and 31771503), Science and Technology Department of Hubei Province of China (CXZD2017000188).

## Author contributions
LQL performed ChIP-Seq; LX, CL, HQX, and HCW verified enhancers and TFs; LQL, ZC, and WCY did bioinformatics, YYL, CM, CN, and YM collected tissues and patient data; WM and LLY directed the project; LQL and WM wrote the paper.

## Competing interests
The authors declare no competing interests.
