## [Peer review file · Nature Communications]

REVIEWER COMMENTS

Reviewer #1 (Remarks to the Author): Expert in ChIP-seq and cancer epigenomics

In this paper, Li et al. use H3K27ac ChIP-Seq in paired colorectal cancer samples to identify active superenhancers, typical enhancers and promoters. They further use dcas9-KRAB to perturb several superenhancers and show that associated genes are downregulated and demonstrate that certain transcription factors, such as KLF3 affect cell migration. This manuscript contains an abundance of epigenomic ChIP-Seq profiling data from patient samples and represents an important resource for the field. The figures are generally well prepared. However, I have several concerns about this current version of the paper.

1. My biggest concern is that the paper seems to be rather fragmented. In the section "the enhancer features of CRC subgroups" the authors say that some enhancers are associated with lipid metabolism. But then no further work is done with lipid metabolism, only cell proliferation and migration assays. In the section on "analysis and verification of variant super enhancer loci" the authors investigate several superenhancers by dcas9-KRAB perturbation (such as VEGFA), but in the next section, they discuss transcription factors, without investigating the effects of the perturbation of these transcription factors on these superenhancers (such as VEGFA).
2. For all siRNA experiments, it seems there was only 1 siRNA used for the non-targeting control. There should be 2 siRNA used, due to off-target effect concerns. Extended Figure S12 - only 1 siRNA was used for siRUNX1 and siMAZ. There should be 2 siRNA.
3. For the ChIP-qPCR, the results should be shown with %input, and the IgG should be shown separately. Sometimes the IgG levels can vary, and this leads to differences that are due to IgG levels.
4. Figure S10D - how are the significances compared? I am not sure what is being compared with what. Is there any idea why P2 is so much lower than P1? Also, the authors should show the genomic distance between the superenhancer and the gene promoter, and include control regions as part of their 3C experiments (the control region is a region between the superenhancer and the gene promoter) and serves as a negative control.
5. For "analysis and verification of variant super enhancer loci", the authors had a section on cell proliferation "data not shown" - this should be shown in the Supplement.
6. To understand the functionality of superenhancers and transcription factors, it is useful to not just perform cell proliferation assays and migration assays but also xenograft assays. This will demonstrate the results seen are not just in vitro but that there is in vivo functionality.
7. In the section on transcription factors, I am confused why the authors discussed ASCL2 as a key TF but then did not select to do experimental verification with ASCL2. Also, did KD affect H3K27ac levels? The data is not shown? Did the KDs affect genes regulated by superenhancers?
8. For the experiments (e.g. Figure 4E) with the dcas9-KRAB, what was the control? I can't find details of how the control was done. The control should be performed as similar to the sgRNA experiment as possible.

Minor concerns:

1. The grammar of this paper reads a bit weirdly and there are some strange phrases. The paper should be proofread.

2. For the figures comparing data e.g. Figure 1F, the results should be compared with random expectation (how much overlap would be seen just by random chance)?

Reviewer #2 (Remarks to the Author): Expert in CRC epigenetics and transcriptomics

Li and co-authors reported an interesting data set of H3K27ac CHIP-Seq on human colorectal cancer. They recruit a relatively large sample repository consist of 73 paired CRC and adjacent normal tissue. Besides H3K27ac CHIP-seq, RNA-Seq, whole genome seq and H3K4me3 CHIP-Seq were also performed on these CRC specimens. In contrast, most previously reported CHIP-seq data of CRC were performed on cell lines, and a few clinic datasets had very limited patient numbers (4 primary CRC reported in Cohen et.al. Nature Comms. 2017, 4 rectal cancer and matched mucosa reported in Flebbe et.al. Cancers 2019). Through the analysis, they identified variant enhancer loci (VEL) specific in tumor tissues then further investigate the VEL in consensus molecular subtypes (CMS). The variant super enhancers loci and transcription factors are also evaluated and validated. Overall, this is a valuable dataset. However, the analysis yielded few insights and some of the conclusions were not substantiated.

Major concerns:

1. This study includes the whole genomic sequencing data. The author should provide more information of mutation data of the patient tumor samples and explore any correlation between mutation and epigenetic states. For example, when analyzing CMS2 VEL, the author mention WNT signaling and cell migration might be APC mutant related, with mutation data, it would be more convincing.
2. The author claims they find 11796 (32.4%) novel enhancers compare to the previous study, are these across all the patients or only some specific patient tumors have these novel enhancers. They also mentioned they ruled out some samples which have the low number of peaks. Based on the Extended Data Fig. S4A, seems only 57 samples were included in the study. Since 16 paired samples (22%) were ruled out, more details and justification need to be provided. The author also needs to provide evidence that the omitting would not introduce the artificial bias (like the bias to PCA analysis: the paper claims H2K4me3 peaks could be used for tumor identification based on PCA. But it is doubtful when more than 1/5 samples were ruled out and only samples with the large number of DE peaks were considered.)
3. The author failed to integrate RNA-seq data with Chip-seq data. How many upregulated DE genes from RNA-seq data are actually caused by Chip-seq identified enhancers? How many of the RNA-seq identified genes have the TF motifs on their promoter/enhancers for the identified TFs?
4. Only migrations attenuated when the author validated VSEL and TF targets, not proliferation. The author's speculation that it was due to the sample being at late/metastatic stages were not convincing. In fact, primary resection are usually early stage tumors and the CMS2 and CMS3 are low EMT related. This may be an artifact of their in vitro culture condition and cell line.
5. The validation of VSEL and TF targets should be performed on patient derived organoids or at least primary patient cancer cells rather than HCT116. Furthermore, for the most important finding such as KLF3, in vivo validations should be considered.
6. The authors decided to validate 4 transcription factors (KLF3, MAFK, MAZ, and RUNX1) that are putatively important for CRC. KLF3 does not show up in 5b as tumor-specific. Additionally, MAFK comes up as healthy-tissue-specific in the same panel. The text body also makes significant mention of ASCL2 as an important hit but is not included for validation. Furthermore, the authors show that knock-down of KLF3 reduces migration in cell lines. However, KLF3 does not show up in 5b as tumor-specific so it is unclear why this factor was chosen for follow-up. Follow-up on the role of KLF3 in the context of CRC is necessary since it doesn't appear to be CRC specific.
7. The patient samples contain not only cancer cells, but also stroma cells. How did the authors distinguish whether the gained and lost enhancers occur in cancer cells or the stroma cells?

Minor concerns:

1. Related to major concern #2, the RNA seq data and MYC promoter and enhancer signal (Figure 1D & 1E) are from all 74 samples or some representatives (like MYC for CMS2).
2. The data that CMS4 has lower common VELs while the VELs per patient are similar, suggesting it is heterogenous over the patients. The CMS4 seems TGF-beta and EMT related, need other information like BRAF and KARS mutation to provide thoughts on the reasons why it is more heterogenous. Could CMS4 be further break-down into more subgroups to get detailed information (or analysis based on individual patients)? On the other hand, the CMS2 is the major subgroup related to MYC and MSS, which may be the reason why it is more homogenous.
3. Page 9 "exhibited attenuated migration ability, including PHF19, LIF, SLC7A5, CYP2S1, RNF43, VEGFA and TBC1D16 (Fig. 4E)." Should be Fig 4F. The staining images correlated with 4F should be shown.
4. The selection criteria of the 4 TFs (KLF3, MAFK, MAZ and RUNX1) needs to be provided.
5. The authors show that their dataset includes samples from several CMS groups, yet the transcription factor analysis is performed on the entire set. I'm curious to know whether the results would be more informative if the analysis was repeated for each molecular subgroup.
6. Also, the HOMER motif analysis was performed on sites of gained and lost signal. Re-running the analysis for sites of gained and lost signal separately would be more informative in understanding tumor- vs healthy-tissue-specific TFs.
7. The authors need to clarify the dCas9-KRAB perturbation assay in more detail. How many enhancers did the author analyze? How many gRNA did the author use for each enhancer? What are the gRNA sequences? The efficiency of the gRNAs needs to be confirmed.

Reviewer #3 (Remarks to the Author): Expert in 3C and epigenomics

In this manuscript, Li et al perform H3K27Ac and H3K4me3 ChIPseq and RNAseq on a large set (43) of pairs of CRC tumors and control tissues. They mostly discuss the H3K27Ac results for 73 pairs of tissues, describe the differences in enhancers found in tumor versus control (Fig.1), describe VELs (variant enhancer loci) and VSELs (variant super enhancer loci) and show that gained VELs preferentially locate near relevant cancer genes (Fig.2), use VELs to identify four CRC sub groups (Fig.3), identify SELs which they silence in HCT116 cells to find (mild) downregulation of nearby genes and a (mild) migratory phenotype of HCT116 cells (Fig.4) and then perform a motif search among gained VELs to identify candidate driver transcription factors. They then characterize one of them, KLF3, a bit further in HCT116 cells.

My comments:

The introduction can be more carefully formulated. For example, the term 'field', used three times, is rather vague. Please have the text be checked by a native speaker, see for example: "Recent studies about aberrant DNA methylation have gain sight of the field" (and many more examples throughout the text).

Further, with respect to the introduction: the emphasis is on the relevance of mapping enhancers/the epigenome in cancers. It would be informative to also get an overview of enhancer mapping exercises in other cancer types (and read what we learned from this), a summary of the different methods to map enhancers (H3K27Ac ChIPseq, RNAseq (eRNAs), ATACseq, and their pros and cons), and get a more detailed picture of previous work on enhancer mapping in CRC (and the conclusions that were drawn). For example, a quick literature search identifies a study by Flebbe et al (Cancers, 2019) in which they similarly mapped enhancers but on a much smaller cohort of primary CRC samples. What do we learn from the current study that is novel over other enhancer mapping studies in cancer/CRC? Finally, it would be good to learn what the most common drivers of CRC are. So authors, please provide a better introduction to this work.

Please mention in the results section the information provided in Fig.S1 and describe the inclusion

criteria of the CRC samples: were there any considerations with regard to age and sex of the patient, the CRC stage, with or without metastasis, whether the genetic drivers was yes/no known? If not, please summarize what sort of samples were included.

Given the large amount of analyzed pairs of samples (73 pairs, 147 H3K27Ac CHIPseq experiments), I would have expected more enhancers to be identified than 27K and 39K in unaffected and tumor tissue. It is difficult to judge the quality of data. Authors, please provide more information on the variability and commonalities between samples. What is the minimum, maximum, median and average number of enhancers scored in control and CRC tissue?

And please also show a correlation matrix and calculate correlation scores: do the control tissues cluster and are they all different from the CRCs? Among the CRCs, can we distinguish sub-groups and do they relate to any of patient parameters summarized in Fig. S1? These metrics are important to judge the quality of the dataset (and to decide how to analyze the data).

Fig.1B: What is summarized here? Is this a merge of all enhancers found across all 'native' and CRC samples? Or do the authors define a threshold, requiring a minimum number of samples showing a given enhancer for it to be included in the analysis? Authors, please make sure this is all clear from the text.

Fig.1C: Is the steepness of the curve indicating large variability between tumor samples? I don't fully understand the meaning of this analysis.

Fig.1D. Please also show for the RNAseq results in a correlation matrix the similarities and differences between samples. Do they cluster similarly as found by ChIPseq? This would nicely confirm that we are analyzing biological, not technical, variation.

"Our RNA-Seq analysis identified 2226 up-regulated different expressed genes (DEGs) and 1979 down-regulated DEGs in CRC tumors (Fig. 1D)." What are the criteria to call DEGs? How is expression variation between control samples and between CRC samples accounted for? And how do these results compare to published RNAseq datasets for CRC?

Fig. 1E. "In tumor tissues, MYC was highly expressed". Is this true for all CRCs? What is the expression variation in control and CRC samples?

Fig.1F: The authors mention that they identify 11796 new enhancers in their 74 CRC samples. They don't mention that they missed 18477 enhancer (uniquely found by Cohen et al in 20 cell lines) and that the overlap (24K) between the two datasets is smaller than the sum of their unique enhancers (30K). A more informative analysis would again be showing a correlation matrix between all samples and the published COAD cell lines ChIPseq results.

In Fig 2 1100 lost and 5590 gained VELs are introduced, determined as discussed in the text and in Fig. S3. Earlier they shown a different analysis (Fig. 1F), mentioning 11K lost and 19K gained enhancers/. What is the value of Fig.1F (of using different H3k27Ac methods and mentioning different numbers of differential H3k27Ac sites?

Please explain in the text how VELs are coupled to genes (linear distance? First neighboring gene?).

Fig.S3F: "the magnitude of the change in expression positively correlated with the number of VELs per gene (Extended Data Fig. S3F).the number of associated gained VELs". Staring at Fig. S3F I wonder: is this significant? The authors may consider checking whether this correlation is better appreciable when considering the (delta) ChIPseq signal strength (ChIP counts) rather than the number of VELs?

Fig.3B See before: please provide such spearman correlation matrix for all samples (CRC and control), based on all enhancers (not just VELs). Do we see the same clusters?

Fig.3E: it is not clear to me whether all VELs were taken here, irrespective of whether they are found in the given subgroup? If so, this does not much over Fig. 3D?

The authors use gene Ontology and a pathway analysis to analyze the genes linked to the VELs for the different subgroups. They conclude for example that subgroup 2genes control WNT signaling and lipid metabolism. While this may be true, they have their RNA-seq data to verify this. Are the associated genes indeed specifically upregulated in the corresponding tumors? Is this true also for other VELs and their associated genes in the other MCS groups?

The authors then repeat the exercise, now focusing on super enhancers (VSELs). This leads to the identification of three (not four) subgroups. Which classification (VEL or VSEL based) should we find most meaningful? Is there any correlation between the four subgroups scored by VELs versus the three found by VSELs?

More generally: the authors promote the importance of considering enhancer profiles to classify tumors. Enhancers control gene expression, of which RNAseq is the direct read-out. When clustering on RNAseq (on differentially expressed genes), are the same four sub-groups identified? If not, what is the value of classifying on enhancers? If yes, what is the added value of classifying on enhancers? The authors perform CRISPRi to silence super enhancers in the HCT116 cell line and show that surrounding genes are downregulated. Is the opposite also true: do the associated genes in the tumor samples show specific upregulation in the corresponding tumors? This seems more relevant information, as we want to know the function of these SVELs (and VELs) in vivo.

The authors probably refer to the wrong figure panel (4E should be 4F?), when stating "The difference of proliferation was not very significant (Data not shown), however, quite a few cell lines exhibited attenuated migration ability, including PHF19, LIF, SLC7A5, CYP2S1, RNF43, VEGFA and TBC1D16 (Fig. 4E)."

In the abstract it says: "Further experiments verified the functions of 6 super enhancers governing PHF19, LIF, SLC7A5, CYP2S1, RNF43 and TBC1D16 in regulating cancer cell migration." I am not an expert on migration assays, but I wonder whether the assay performed on a single cell line HCT116, and the results shown in Fig. 4F, are sufficient to draw this conclusion.

The 3C data are not essential for the paper, in my mind, but are also not of sufficient quality and, as presented, don't allow drawing conclusions on looping. 3C is an very difficult to control assay: each given ligation product is extremely rare and consequently, accurate quantification of such rare products is very difficult. The control used (fragment-internal primerpair) correct for differences in amounts of template, but not for differences in crosslinking, digestion and ligation. More importantly, there is no correction carried out for differences in amplification efficiencies between primer pairs, which strongly impact results. Having a control template with equal amounts of all analyzed ligation products, to determine for each pair its amplification efficiency and correct for this, is crucial for correct interpretation. Finally, results need to be plotted on top of the loci, such that the chromosomal distances between analyzed fragments become appreciable. In B (CEBPB), provided that results after normalization will look the same (which I predict is unlikely) it is unclear why P3 and P4 (both on the enhancer) are so different in contact. In C (VEGFA) it currently is strange to see such big differences between P3 and P4 (that are extremely close on the chromosome). In D (CYP2S1) the most distal fragment P1, which seems upstream of the enhancer, appears to loop to the gene: why. In C and D, to appreciate a loop, you would want to analyze two fragments in between the gene and enhancer. Fig.5B (and C): "Heatmap of transcription factors ranked by predicted core regulatory circuitry (CRC) total degrees (Tumor - Native tissue). Top 30 tumor and native specific TFs were listed." Please explain in the text what this analysis entails.

It states: "the gene expression analysis based on TCGA datasets suggested ASCL2 was highly expressed specific in colorectal cancer (Extended Data Fig. S11E)." Why based on TCGA? The authors generated their own RNAseq data and should discuss how ASCL2, and the other identified factors shown in Fig. 5C, express in their own control and CRC tissues. Are they upregulated in their CRCs? If not, it is difficult to understand how they would suddenly act as drivers.

The added value of mapping enhancers in tumors, as compared to a transcriptome analysis, to me seems not to come from an improved ability to classify tumors (discussed above), but indeed to learn from the VELs and VSELs the underlying motifs, which can help identifying driver transcription factors. Only Figure 5 is dedicated to this exercise. The top hit, ASCL2 (Fig.5C), is mentioned but ignored in follow up studies and instead four TFs are selected (RUNX1, MAZ, MAFK and KLF3) for a limited analysis (proliferation and migration assay upon knockdown in one cell line). These selected factors, curiously, don't show up from their own ChIP-seq analysis (Fig.5C), but were all (except for RUNX1) selected from literature. This twist is hard to understand. If the authors want to make the point (as they do) that systematic enhancer mapping is important to understand the biology of tumors, they should seriously characterize their top hit TFs and show that indeed these are key factors in tumorigenesis.

The abstract claims: "we identified KLF3 as a novel oncogenic transcription factor in CRC." I don't think the limited data (Fig.5D-F) on a cell line allow drawing this conclusion. Also, it is unclear where KLF3 suddenly comes from: it does not show up among the top candidates (Fig.5C), and the KLF motif is hardly enriched (Fig. 5A).

Reviewer #4 (Remarks to the Author): Expert in CRC genomics and whole-genome sequencing

Li et al present an impressive amount of data which could really improve our current understanding of the mechanisms underlying CRC. However, I feel the current way of analysing the data could be improved and the RNAseq data could be used more efficiently.

Specific comments:

1) ChIP-seq analysis:

- Why were paired tumor and normal ChIP-seq data not used together in the MACS2 pipeline? Normal samples could be used as the control sample which would automatically yield a list of differential peaks between tumor and normal for every patient. From this list the authors could extract VELs and VSELs per patient and determine their recurrence in the total cohort.

To me this seems a more elegant and statistically sound approach than the one the authors currently use, in which arbitrary and debatable cut offs for fold changes are used to define VELs and VSELs.

- The authors indicate multiple times that patients were excluded from certain analyses because low numbers of VELs or VSELs were found. However, when the data generated passes the quality control (so sufficient numbers of unique reads, good phred scores etc) they cannot be excluded from further analysis for the simple reason that their results (from the comparison tumor vs normal) are not what the authors were expecting. The reasoning that this may represent a sampling problem may be true but cannot be made conclusive when no HE slides have been evaluated for the tissues.

- CMS specific VELs should be identified by a statistical comparison between the H3K27Ac signal between CMS groups, instead of only looking at a fold change.

2) RNAseq analysis:

- The authors state that 20M reads were generated for every sample, however I assume that also here there will be a range (similar to the 30-50M reads for ChIP-seq). Please indicate this in the relevant section.

- DEGs were identified using DESeq2. Was a paired t-test used for this? The authors write they used contrast adjustment for multiple groups comparison. I think this indicates that they used the multifactor setting to control for the patient effect, which effectively comes down to a paired analysis. Could the authors please clarify this?

- CMS calling: the authors need to indicate in the manuscript how many CMS1, CMS2, CMS3, CMS4 and no group samples were identified in their cohort (this is now only given in the supplementary figures). This is important for readers to judge whether CMS2 is really the only group showing specific VELs or whether this is due to sample numbers (since CMS2 is the largest group). In addition, I feel that the "no group" samples should still be included in the analysis and it seems they are currently left out?

- As far as I can tell the RNAseq data was not used to verify whether the identified VELs and VSELs actually have an effect on the expression of their associated SE-genes. This analysis would really add to the manuscript as it would demonstrate the effect of the differentially activated V(S)ELs in CRC.

3) General remarks:

- In the cluster analysis identifying 3 CRC subgroups it would be interesting if the CMS group would also indicated per sample to see whether the subgroups based in V(S)ELs are overlapping with the CMS subgroups or not.
- Loss of KLF3 was previously shown to be associated with more aggressive CRC (PMID: 28423541). This seems to be somewhat contradictory to the current findings. Could the authors comment on that?

To whom it may concern,

Thank you for giving us a chance to submit our revised manuscript, titled “*Genome-wide profiling of active enhancers in colorectal cancer*” (NCOMMS-20-47734-T). We have performed additional experiments to address the comments raised by the reviewers. Below is our point-by-point response to each comment, in which the text in blue is the original comments from the reviewers and our response in black.

Reviewer #1 (Remarks to the Author): Expert in ChIP-seq and cancer epigenomics

In this paper, Li et al. use H3K27ac ChIP-Seq in paired colorectal cancer samples to identify active superenhancers, typical enhancers and promoters. They further use dcas9-KRAB to perturb several superenhancers and show that associated genes are downregulated and demonstrate that certain transcription factors, such as KLF3 affect cell migration. This manuscript contains an abundance of epigenomic ChIP-Seq profiling data from patient samples and represents an important resource for the field. The figures are generally well prepared. However, I have several concerns about this current version of the paper.

Response: We appreciate the reviewer for his/her positive comment.

1. My biggest concern is that the paper seems to be rather fragmented. In the section "the enhancer features of CRC subgroups" the authors say that some enhancers are associated with lipid metabolism. But then no further work is done with lipid metabolism, only cell proliferation and migration assays. In the section on "analysis and verification of variant super enhancer loci" the authors investigate several superenhancers by dcas9-KRAB perturbation (such as VEGFA), but in the next section, they discuss transcription factors, without investigating the effects of the perturbation of these transcription factors on these superenhancers (such as VEGFA).

Response: The current study is much more like a resource study, so we did not put much effort in mechanistic studies. We investigated the distribution of CRC-specific enhancers, analyzed their potential use in classification, predicted novel regulators and then verified experimentally. Several studies have actually already revealed the connection between lipid metabolism and CRC¹, so we did not go further with it. Our study predicted multiple super enhancers and transcription factors functioning in CRC. We then validated 11 super enhancers and identified 10 of them that regulate target gene expression, most of which regulate cell migration (Fig.4). To further solidify our results, we performed more experiments in multiple cell lines and animal models during revision. Our new data showed that silencing of the above super enhancers has similar effects in RKO and SW620 cell lines; and silencing of super enhancers for PHF19 and TBC1D16 significantly repressed tumor formation in mice (Fig. 4G-I, Extended Data Fig. S11). For TFs, we identified KLF3 as a novel oncogenic gene and also confirmed with a series of experiments (Fig. 5D-H, Extended Data Fig. S14). All the work is closely related with our epigenomic study in enhancer biology in CRC, and is useful to the potential usage of our data.

2. For all siRNA experiments, it seems there was only 1 siRNA used for the non-targeting control.

There should be 2 siRNA used, due to off-target effect concerns. Extended Figure S12 - only 1 siRNA was used for siRUNX1 and siMAZ. There should be 2 siRNA.

Response: For MAZ and RUNX1, their functions on CRC cells have been published already, so after the first-round screen and when we found the results were similar to the published, we did not focus on these genes. For MAFK, two siRNAs were used although it seemed to have no function in CRC (Extended Data Fig. S13). Then we focused on KLF3. Knockdown with two different siRNAs significantly repressed cell migration (Extended Data Fig. S14A&B). During revision, we performed knockdown assays with two different sgRNAs in two cell lines, and measured cell proliferation, migration and tumor formation with xenograft. All the results agreed the previous conclusion, that KLF3 deficiency repressed cell migration and tumor formation, indicating an oncogenic role for KLF3 (Extended Data Fig. S14C-H, Fig. 5D-G).

3. For the ChIP-qPCR, the results should be shown with %input, and the IgG should be shown separately. Sometimes the IgG levels can vary, and this leads to differences that are due to IgG levels.

Response: The result normalized with input was shown as below (Response Fig. 1). We used that normalized with IgG because we think IgG is a nice control for antibodies. If the reviewer think it is necessary to replace the figure in the manuscript, we can do the replacement.

Response Fig. 1 H3K27ac enrichment on genes targeted by sgRNAs and dCas9-KRAB.

4. Figure S10D - how are the significances compared? I am not sure what is being compared with what. Is there any idea why P2 is so much lower than P1? Also, the authors should show the genomic distance between the superenhancer and the gene promoter, and include control regions

as part of their 3C experiments (the control region is a region between the superenhancer and the gene promoter) and serves as a negative control.

Response: In 3C, we are comparing the quantitative PCR results with a primer pair on *GAPDH* gene without DpnII cut sites, which is commonly used in many 3C studies and represents the background signal of the experiment. We missed the information in the old version, and now we have described it in the Method section and added the sequence in the supplemental table. We also agree that 3C experiments sometimes is confusing because of the variation of enzyme digestion sites and PCR efficiency. Here, we just want to show that we could get positive PCR results between two faraway loci, which supports that the two fragments of DNA are spatially close.

5. For "analysis and verification of variant super enhancer loci", the authors had a section on cell proliferation "data not shown" - this should be shown in the Supplement.

Response: We have added the results to Extended Data. Fig. S11B.

6. To understand the functionality of superenhancers and transcription factors, it is useful to not just perform cell proliferation assays and migration assays but also xenograft assays. This will demonstrate the results seen are not just in vitro but that there is in vivo functionality.

Response: We have performed xenograft experiments during revision. Our data showed that repression of super enhancers for PHF19 and TBC1D16, and deficiency of KLF3, significantly repressed tumor formation in mice (Fig. 4G-I, Fig. 5F-H).

7. In the section on transcription factors, I am confused why the authors discussed ASCL2 as a key TF but then did not select to do experimental verification with ASCL2. Also, did KD affect H3K27ac levels? The data is not shown? Did the KDs affect genes regulated by superenhancers?

Response: Several studies have well established the role of ASCL2 in CRC^{2,3}, so we used it as a representative to support our analysis. Since KLF3 has not been reported in CRC, we focused on it and proved that it is an oncogenic TF in CRC.

8. For the experiments (e.g. Figure 4E) with the dCas9-KRAB, what was the control? I can't find details of how the control was done. The control should be performed as similar to the sgRNA experiment as possible.

Response: In Fig. 4E, we used a sgRNA targeting EGFP as control in all experiments. We have clarified it in the figure legend in the revised manuscript.

Minor concerns:

1. The grammar of this paper reads a bit weirdly and there are some strange phrases. The paper should be proofread.

Response: We have gone through the whole manuscript and asked one native speaker to correct

for us.

2. For the figures comparing data e.g. Figure 1F, the results should be compared with random expectation (how much overlap would be seen just by random chance)?

Response: To address your concern, we randomly selected the same number of regions in human genome, whose length are also the same as the significant enhancers called from 20 CRC cell lines data, and repeated it three times. The result showed that when we selected the same coverage of the human genome randomly, the number of overlapped regions with significant CRC patient enhancers is around 10 times less than cell line data (Response Fig. 2). It indicates that the relevance between our data and published cell line data is significant.

Response Fig. 2 Bar plot shows that enhancers from CRC cell line data have about 10 times higher chance of overlapping with CRC patient enhancers than random regions. Random regions have the same length as CRC cell lines identified enhancers.

Reviewer #2 (Remarks to the Author): Expert in CRC epigenetics and transcriptomics

Li and co-authors reported an interesting data set of H3K27ac CHIP-Seq on human colorectal cancer. They recruit a relatively large sample repository consist of 73 paired CRC and adjacent normal tissue. Besides H3K27ac CHIP-seq, RNA-Seq, whole genome seq and H3K4me3 CHIP-Seq were also performed on these CRC specimens. In contrast, most previously reported CHIP-seq data of CRC were performed on cell lines, and a few clinic datasets had very limited patient numbers (4 primary CRC reported in Cohen et.al. Nature Comms. 2017, 4 rectal cancer and matched mucosa reported in Flebbe et.al. Cancers 2019). Through the analysis, they identified variant enhancer loci (VEL) specific in tumor tissues then further investigate the VEL in consensus molecular subtypes (CMS). The variant super enhancers loci and transcription factors are also evaluated and validated.

Overall, this is a valuable dataset. However, the analysis yielded few insights and some of the conclusions were not substantiated.

Response: We appreciate the reviewer for his/her positive comments. During revision, we have tried our best to address all the questions. I hope it will be satisfying.

Major concerns:

1. This study includes the whole genomic sequencing data. The author should provide more information of mutation data of the patient tumor samples and explore any correlation between mutation and epigenetic states. For example, when analyzing CMS2 VEL, the author mention WNT signaling and cell migration might be APC mutant related, with mutation data, it would be more convincing.

Response: I agree with the reviewer it is more meaningful to combine the genomic and epigenetic information in our dataset. Actually, we are further analyzing our data based on the similar concept after we submitted our manuscript. To address the above question, we utilized the software GATK4 to find mutations in APC coding region among all patients. We totally identified 19 APC mutated patients in 57 patients who contained in CMS subgroups (33.3%). Based on the statistical data, we did not see the significant correlation between CMS2 and APC mutation (Response Fig. 3). We will further work on it and probably report the result in our future study.

	Number	Propotion in specific subgroup
CMS1	5	0.50
CMS2	6	0.32
CMS3	3	0.23
CMS4	5	0.33

Response Fig. 3 Chart for the number and the proportion of APC mutation patient in specific CMS subgroups.

2. The author claims they find 11796 (32.4%) novel enhancers compare to the previous study, are these across all the patients or only some specific patient tumors have these novel enhancers. They also mentioned they ruled out some samples which have the low number of peaks. Based on the Extended Data Fig. S4A, seems only 57 samples were included in the study. Since 16 paired samples (22%) were ruled out, more details and justification need to be provided. The author also needs to provide evidence that the omitting would not introduce the artificial bias (like the bias to PCA analysis: the paper claims H2K4me3 peaks could be used for tumor identification based on PCA. But it is doubtful when more than 1/5 samples were ruled out and only samples with the large number of DE peaks were considered.)

Response: we answer the above questions one by one.

- 1) The criterion we used for identifying significant enhancers is that they should appear in at least two patients. For most of these novel enhancers, only a portion of patient tumors have them. We provide a plot below to describe the relationship between the number of novel enhancer and their frequency in patients here (Response Fig. 4).

Response Fig. 4 Dot plot describing the relationship between the number of novel CRC enhancer and their appearance in patients.

- 2) Actually, there are 7 patient tumor samples (patient #3, #32, #35, #47, #68, #72, #73) have not been grouped according to the results from CMScaller software (it's quite common when doing CMS classification), so we didn't show this part of patient identifiers in Extended Data Fig. S4A. For the other 9 samples, and we have explained the reason why we did not use them in the Method (section "ChIP-seq data processing" and "Identification of VELs"). These 9 samples consist of two parts. 4 samples (patient #20, #21, #22, #24) were deleted because of their low number of significant H3K27ac peaks (Extended Data Fig. S3A). Based on our experience and some published papers, we set 2500 as the number threshold, and we could see that the peak numbers of these 4 samples are less than 2500, which was maybe due to low quality of ChIP assay during sample preparation. We checked our H3K27ac ChIP-seq results for the 4 samples, including the concentration of the ChIP DNA and the sequencing reads number. We found the prepared samples of native tissues of the 4 pairs had less DNA than others, which was probably due to less cell number than other tissues, which is often a nightmare for ChIP assay with patient tissues. Their paired tumor data is normal, and if we keep these 4 patients for VEL identifying, we'll get lots of false candidates, which is not good for our analysis.

The reason for ruling out the other 5 patients (patient #8, #19, #38, #41, #57) in VEL identifying is that there is only little difference between the paired tissues in individual VEL number. We mentioned in the manuscript that the first step we calling VEL is to identify all VELs for every patient, the cut-off is Fold Change > 2. For individual patient, if their VEL number (GAIN VEL + LOST VEL) is less than 500, we treated them as the "No difference samples" (Extended Data Fig. S3B). The purpose of identifying VEL is to find the difference between tumor and native tissue, so these "No difference samples" don't play a role in this process. If we keep these samples in further process, it may influence the step of statistical test (calculating padj and recurrence). Based on these reasons, we decided to exclude these 5 patients in VEL identifying process.

- 3) The author failed to integrate RNA-seq data with Chip-seq data. How many upregulated DE genes from RNA-seq data are actually caused by Chip-seq identified enhancers? How many of the RNA-seq identified genes have the TF motifs on their promoter/enhancers for the identified TFs?

Response: We did perform the integrated analysis but did not show the data in the original manuscript, because the data amount is very large and it would not provide much information if we just show some statistical results. So, we just showed the corresponding RNA-seq data when we need to display some examples, such as Fig. 1E and Extended Data Fig. S6I-L. Now we have added relevant data to the revised manuscript.

To answer your first question, we counted the number of significant enhancers near DEGs, including enhancers for up-regulated DEGs in tumor and native tissues, respectively (Response Fig. 5A). We did not identify proximal active enhancers for 50% up-regulated DEGs in tumor and 37% up-regulated DEGs in native tissues. It is probably because of two reasons. One is that we can't determine the accurate relationship between genes and enhancers, and the enhancers are outside of the regions we searched. It will be much better if 3d genome information is available. The other is that the identification of CRC significant enhancers and DEGs are based on statistical results, so it's too hard to use these two statistical results to get accurate results. Nevertheless, according to the result, around 41% up-regulated DEGs in tumor and 37% up-regulated DEGs in native tissues were caused by the ChIP-seq identified enhancers, respectively (Response Fig. 5B). We have added the figures into Extended Data as Fig. S2F&G.

Response Fig. 5 Correlation between DEGs and enhancers in the adjacent and tumor tissues. (A) Dot plot shows the relationship between the gene number and associated enhancer number in tumor and native tissue. (B) Bar plot for the proportion of DEGs with different enhancer signal alteration in tumor and native tissues.

About TF motifs, it's too hard to analyze the binding of all TFs, and we just chose 5 TFs mentioned in our manuscript (ASCL2, KLF3, MAFK, MAZ and RUNX1) for analysis. We integrated the motif position information from all samples and corresponded them to the promoter and enhancer regions of all DEGs. If one motif of one specific TF appears more than once in a DEG's promoter or enhancer, we treated this DEG as the specific TF bound DEG. We analyzed all these 5 TFs, and the results showed that the proportion of DEGs which bound by specific TF fluctuates around 30% (Response Fig. 6). It seemed that no significant difference exists between TF-regulated DEGs in tumor and native tissues. However, such bioinformatic analysis is not accurate, and it will be much better to determine TF target genes by performing RNA-Seq and

ChIP-Seq analyses in the same cells.

Response Fig. 6 Proportion of DEGs bound by 5 transcription factors (ASCL2, KLF3, MAFK, MAZ and RUNX1). Transcription factor binding positions were obtained from [crc](https://github.com/linlabcode/CRC) (<https://github.com/linlabcode/CRC>).

- 4) Only migrations attenuated when the author validated VSEL and TF targets, not proliferation. The author's speculation that it was due to the sample being at late/metastatic stages were not convincing. In fact, primary resection are usually early stage tumors and the CMS2 and CMS3 are low EMT related. This may be an artifact of their in vitro culture condition and cell line.

Response: We agreed that it is possibly caused by our in vitro culture condition and cell line. However, we have repeated the assays in multiple cell lines during revision and obtained similar results. Meanwhile, in the same system, we observed difference of cell proliferation in other studies. So, the possibility is quite small. Other possibilities could be that since the normal intestine epithelial cells are growing rapidly, the difference between migration-related genes in tumor and native tissues is perhaps bigger than proliferation genes, which caused the top candidates are mostly related with migration. We have discussed the possibility in the revised manuscript.

- 5) The validation of VSEL and TF targets should be performed on patient derived organoids or at least primary patient cancer cells rather than HCT116. Furthermore, for the most important finding such as KLF3, in vivo validations should be considered.

Response: We agree that it will be much better to validated the functions of VSELS and TFs in patient-derived organoids or cells. However, we encountered difficulty to obtain patient tissues because of COVID crisis. Then we chose to perform validation in more CRC cell lines and xenograft experiments in mice. As shown in Fig 4G-I & 5F-H, Extended Data Fig. S11&14 in the revised manuscript, we have tried our best to provide data to address the question, and hope the reviewer could understand our situation.

- 6) The authors decided to validate 4 transcription factors (KLF3, MAFK, MAZ, and RUNX1) that are putatively important for CRC. KLF3 does not show up in 5b as tumor-specific.

Additionally, MAFK comes up as healthy-tissue-specific in the same panel. The text body also makes significant mention of ASCL2 as an important hit but is not included for validation. Furthermore, the authors show that knock-down of KLF3 reduces migration in cell lines. However, KLF3 does not show up in 5b as tumor-specific so it is unclear why this factor was chosen for follow-up. Follow-up on the role of KLF3 in the context of CRC is necessary since it doesn't appear to be CRC specific.

Response: As we mentioned in the manuscript, we did not follow up with ASCL2 because a number of studies had already shown its function in CRC²⁻⁴. We just showed the analyzed data in the supplemental materials.

To increase the number of predicted candidates, we used two approaches for TF prediction. Some novel candidates did not pop up in both analysis, such as KLF3. The bioinformatic analysis were used for TF prediction and its accuracy was highly dependent on the available data. For some TFs, many published ChIP-Seq data are available, but others only have very few data. So, although some TFs were predicted by one approach, it is still possible that they play important roles in CRC. We chose 10 predicted TFs for validation, but some genes express extremely low in the selected cell line or knockdown experiments did not work with some siRNAs, so we just presented data of the mentioned 4 TFs. We have clarified it in the revised manuscript.

7. The patient samples contain not only cancer cells, but also stroma cells. How did the authors distinguish whether the gained and lost enhancers occur in cancer cells or the stroma cells?

Response: It is a very good question. It will be much better to isolate specific cells and then perform epigenomic study. But the current technique is still very limited, especially when handling a large number of tissues. Technically, it is possible to do STAR-ChIP or CUT&TAG after isolating specific cells with flow cytometry. But till now, we have not seen such publication yet. The samples we used were a mixture of cancer and other cells. Based on our RNA-Seq and ChIP-Seq results, we could find typical oncogenes in DEGs and predict oncogenic TFs reported in previous publications, which means our data are still representable for CRC tissues.

Minor concerns:

1. Related to major concern #2, the RNA seq data and MYC promoter and enhancer signal (Figure 1D & 1E) are from all 74 samples or some representatives (like MYC for CMS2).

Response: The result shown in Fig. 1D are generated from the RNA-seq data of all samples, more detailed information could be found in Method "RNA-seq data processing and DEG identification" section.

For Fig. 1E, we randomly choose 10 patients to draw this plot (not just CMS2). We repeated this analysis using all samples, and the result has shown below (Response Fig. 7). It has almost the same tendency as Fig. 1E.

Response Fig. 7 Normalized ChIP-seq and RNA-seq Meta tracks showing H3K27ac and mRNA signal on MYC promoter and enhancer loci, related signals are averaged by all samples.

2. The data that CMS4 has lower common VELs while the VELs per patient are similar, suggesting it is heterogenous over the patients. The CMS4 seems TGF-beta and EMT related, need other information like BRAF and KARS mutation to provide thoughts on the reasons why it is more heterogenous. Could CMS4 be further break-down into more subgroups to get detailed information (or analysis based on individual patients)? On the other hand, the CMS2 is the major subgroup related to MYC and MSS, which may be the reason why it is more homogenous.

Response: Thanks for your suggestions. The research goal for this part is to recapitulate the traits of enhancers signal in different CMS subgroups, and to further classify CMS4 group into more subgroups is out of our purpose. CMS is a clinical-based classification, so the enhancer landscape heterogenous phenomenon in CMS4 is reasonable. It requires more information and much larger samples size to determine the standard for CMS4 subgroups, which is out of the goal of the current study.

About CMS2, we agree that its signatures are better determined and related with known genetic factors, which may be the reason for its homogeneity.

3. Page 9 “exhibited attenuated migration ability, including PHF19, LIF, SLC7A5, CYP2S1, RNF43, VEGFA and TBC1D16 (Fig. 4E).” Should be Fig 4F. The staining images correlated with 4F should be shown.

Response: We have made the correction in the revised manuscript, and added some images of cell migration assays into the Extended Data Fig. S11 C&D.

4. The selection criteria of the 4 TFs (KLF3, MAFK, MAZ and RUNX1) needs to be provided.

Response: We have described it in the revised version. We actually chose 10 TFs when initiating the validation. However, some express extremely low in the selected cells, or siRNAs did not work during the first-round experiment. Eventually, we only got successful knockdown of 4 TFs. Among them, MAZ and RUNX1 have been recently reported as the promoter for COAD development, and they also appeared in our list (Top 50 p-value for MAZ in Fig. S11B, red highlighted; tumor specific rank 26 for RUNX1 in Fig. 5B). Our results support the previous publications nicely and they could be treated as positive controls.

KLF3 and *MAFK* are not reported yet. One case study has investigated *KLF3* expression in CRC tissues⁵, but no experimental studies were found. They appeared in Fig. S12B and can be significantly predicted in some individual cases (Extended Data Fig. S12F). So, we chose them for identifying some new TFs playing the key role in COAD development.

- The authors show that their dataset includes samples from several CMS groups, yet the transcription factor analysis is performed on the entire set. I'm curious to know whether the results would be more informative if the analysis was repeated for each molecular subgroup.

Response: We have performed the suggested analysis (Response Fig. 8). We first picked out the top 30 tumor specific TFs of all CMS subgroups based on their total CRC degree (described in Fig. 5B), and then merged all 4 lists together (n = 66). We draw a heatmap to illustrate the difference between all these subgroups, and we could see all TFs were divided into 3 parts. The top and middle parts represented the TFs with similar CRC degree in all CMS subgroups. The bottom part (highlighted by the black box) included the TFs with distinguished CRC degree between these subgroups. For instance, *HNF4A*, *PLAGL2*, *EHF*, *KLF5* and *VDR* had the significant higher degree in CMS2 than the other 3 subgroups. So, we concluded that most of the TFs have the similar situation in all CMS subgroups, but around 1/3 TFs have the subgroup-specific traits. It is interesting because it is possible a new strategy to use the subgroup-specific TFs to classify cancer subgroups.

Response Fig. 7 Heatmap for 66 transcription factors' total CRC degree in 4 CMS subgroups.

- Also, the HOMER motif analysis was performed on sites of gained and lost signal. Re-

running the analysis for sites of gained and lost signal separately would be more informative in understanding tumor- vs healthy-tissue-specific TFs.

Response: Actually, we analyzed these two parts of VEL separately, exactly as your advice. The HOMER motif result from gain VELs was shown in Fig. 5A, and result from lost VELs was shown in Extended Data Fig. S12A.

7. The authors need to clarify the dCas9-KRAB perturbation assay in more detail. How many enhancers did the author analyze? How many gRNA did the author use for each enhancer? What are the gRNA sequences? The efficiency of the gRNAs needs to be confirmed.

Response: Totally we analyzed 11 SEs, among which 10 of them were found to regulate the expression of their proximal genes. We have added the information into the revised manuscript (page 9, line 227-230). The site of each sgRNA was shown in the illustration of each SE and the sequences for sgRNAs were shown in Sup. Table 8. To ensure the effect of inhibition, we transfect multiple sgRNAs together into cells, the results shown are the phenotypes for the mixtures of pooled cells. We examined the efficiency of enhancer inhibition by measuring H3K27ac on enhancers and target gene expression (Extended Data Fig. S9&10A, Response Fig. 1).

Reviewer #3 (Remarks to the Author): Expert in 3C and epigenomics

In this manuscript, Li et al perform H3K27Ac and H3K4me3 ChIPseq and RNAseq on a large set (43) of pairs of CRC tumors and control tissues. They mostly discuss the H3K27Ac results for 73 pairs of tissues, describe the differences in enhancers found in tumor versus control (Fig.1), describe VELs (variant enhancer loci) and VSELs (variant super enhancer loci) and show that gained VELs preferentially locate near relevant cancer genes (Fig.2), use VELs to identify four CRC sub groups (Fig.3), identify SELs which they silence in HCT116 cells to find (mild) downregulation of nearby genes and a (mild) migratory phenotype of HCT116 cells (Fig.4) and then perform a motif search among gained VELs to identify candidate driver transcription factors. They then characterize one of them, KLF3, a bit further in HCT116 cells.

My comments:

The introduction can be more carefully formulated. For example, the term ‘field’, used three times, is rather vague. Please have the text be checked by a native speaker, see for example: “Recent studies about aberrant DNA methylation have gain sight of the field” (and many more examples throughout the text).

Response: We have gone through the whole manuscript and asked one native English speaker to check it for us.

Further, with respect to the introduction: the emphasis is on the relevance of mapping enhancers/the epigenome in cancers. It would be informative to also get an overview of enhancer mapping exercises in other cancer types (and read what we learned from this), a summary of the different methods to map enhancers (H3K27Ac ChIPseq, RNAseq (eRNAs), ATACseq, and their

pros and cons), and get a more detailed picture of previous work on enhancer mapping in CRC (and the conclusions that were drawn). For example, a quick literature search identifies a study by Flebbe et al (Cancers, 2019) in which they similarly mapped enhancers but on a much smaller cohort of primary CRC samples. What do we learn from the current study that is novel over other enhancer mapping studies in cancer/CRC? Finally, it would be good to learn what the most common drivers of CRC are. So authors, please provide a better introduction to this work.

Response: We have re-written the introduction and added the mentioned literature. Please refer to the introduction part of the revised manuscript.

Please mention in the results section the information provided in Fig.S1 and describe the inclusion criteria of the CRC samples: were there any considerations with regard to age and sex of the patient, the CRC stage, with or without metastasis, whether the genetic drivers was yes/no known? If not, please summarize what sort of samples were included.

Response: The tissues were collected from patients who received surgical treatment at Zhongnan Hospital of Wuhan University (Wuhan, China) between August 2017 and February 2018. The patients were mostly from the Huazhong area of China, especially Hubei province. To gain a full picture of CRC epigenome, no specific criteria was applied when collecting tissues. The tumor samples and patient information was summarized in the Supplemental Table 1. We have also added more description in the results.

Given the large amount of analyzed pairs of samples (73 pairs, 147 H3K27Ac CHIPseq experiments), I would have expected more enhancers to be identified than 27K and 39K in unaffected and tumor tissue. It is difficult to judge the quality of data. Authors, please provide more information on the variability and commonalities between samples. What is the minimum, maximum, median and average number of enhancers scored in control and CRC tissue?

Response: We have provided the corresponding information about all NGS data reads number and all ChIP-seq data peak number in Supplementary Table 2 “Information of all sequencing data”, from which we can see that the read numbers for all samples are sufficient for H3K27ac ChIP-seq. For the data quality, we calculated Q20 and Q30 (forward 120bp, because we cut the terminal 30bp for all reads) for all H3K27ac ChIP-seq data (Response Fig. 8). The result shows that for most of the samples, base Q20 are larger than 90%, and Q30 percentage are larger than 80%, indicating that our H3K27ac ChIP-seq data quality is good.

Response Fig. 8 Sequencing data quality (Q20 and Q30) of H3K27ac ChIP-seq samples (divided by different paired end data R1 and R2) from all tumor and native tissues.

The criterion we used for identifying significant enhancers is that they should appear in at least two samples. All 27K and 39K enhancers we showed in Fig. S2A meet the standard. If we consider enhancers identified in all samples, the number would be 44K for native tissues and 67K for tumors.

We have shown H3K27ac peak numbers for all samples in Extended Data Fig. S3A. For the enhancer numbers in all individual samples, the minimum, maximum, mean and median numbers are shown below (Response Fig. 9). Some native tissue samples with very low enhancer number in (e.g. N20, N21, N22 and N24) were removed in our further analysis.

Response Fig. 9 Dot plot (left panel) and statistical values (right panel) of Enhancer number for all the CRC patients native and tumor samples.

And please also show a correlation matrix and calculate correlation scores: do the control tissues cluster and are they all different from the CRCs? Among the CRCs, can we distinguish sub-groups and do they relate to any of patient parameters summarized in Fig. S1? These metrics are important to judge the quality of the dataset (and to decide how to analyze the data).

Response: We merged all native and tumor significant enhancers we mentioned before as the reference region, then calculated the enhancer signal (H3K27ac RPM) correlation score for all samples and generated the corresponding matrix (Response Fig. 10). The result showed that most of the enhancer profiles can be divided into two obvious clusters according to their tissue types. So, we can say that the enhancer profile of native tissues is different from tumor tissues, which is consistent with our PCA result.

Actually, we analyzed the clinical information of patients, and tried to find the correlation with CMS classified sub-group identifiers. Unfortunately, we didn't find obvious connections between the two sets of information, so we didn't show this part in our manuscript.

Response Fig. 10 Heatmap for the spearman correlation of enhance signal (H3K27ac RPM) among tumor and native tissue samples from all CRC patients.

Fig.1B: What is summarized here? Is this a merge of all enhancers found across all ‘native’ and CRC samples? Or do the authors define a threshold, requiring a minimum number of samples showing a given enhancer for it to be included in the analysis? Authors, please make sure this is all clear from the text.

Response: The enhancers used here are the native and tumor significant enhancers we mentioned in the above question. They are the enhancers appeared at least twice in all native or tumor samples. We have added the description in page 5, line 130-133.

Fig.1C: Is the steepness of the curve indicating large variability between tumor samples? I don’t fully understand the meaning of this analysis.

Response: Fig. 1C shows a saturation analysis result which is used to make sure our sample size is sufficient to cover most CRC enhancers. Briefly, when we increase the number of samples, we will get some new enhancers from the newly added samples. As we keeping it increasing, the sum of enhancers will get into a plateau, which means newly added samples can’t provide more new enhancers. In Fig. 1C, we can see in the first 20 samples, the curve is very steep; and after 50 samples, the curve tends to be flat and these 50 samples could cover 90% CRC enhancer generated from all tumor samples. So, this result indicated us that our sample size is sufficient for enhancer analysis.

Fig.1D. Please also show for the RNAseq results in a correlation matrix the similarities and differences between samples. Do they cluster similarly as found by ChIPseq? This would nicely confirm that we are analyzing biological, not technical, variation.

Response: We used the gene expression profiles of all samples to generate the correlation matrix, and the result is shown below (Response Fig. 11). As the correlation matrix in Response Fig. 10, CRC patient gene expression profiles can also be well divided in two parts. It is also consistent with our PCA result, and we can conclude that enhancer and gene expression profiles of CRC patients could be used for distinguishing the tumor and native tissues.

Response Fig. 11 Heatmap for the spearman correlation of gene expression profiles among tumor and native tissues from all CRC patients.

“Our RNA-Seq analysis identified 2226 up-regulated different expressed genes (DEGs) and 1979 down-regulated DEGs in CRC tumors (Fig. 1D).” What are the criteria to call DEGs? How is expression variation between control samples and between CRC samples accounted for? And how do these results compare to published RNAseq datasets for CRC?

Response: For the criteria to call DEGs, we have described in the Method “RNA-seq data processing and DEG identification” section. Briefly, we utilized the R package DESeq2 to identify DEGs with the cut-off of $\log_2FC > 1$ and $p_{adj} < 0.01$. The expression variations between control samples was expected smaller than tumor tissues, as shown in the correlation analysis (Response Fig. 11). For the public CRC RNA-seq data from TCGA, we identified their DEGs by the same criteria we mentioned above. Then we used the DEG lists of these two datasets to identify overlapped genes and generated the Venn plot of Fig. S2D.

Fig. 1E. “In tumor tissues, MYC was highly expressed”. Is this true for all CRCs? What is the expression variation in control and CRC samples?

Response: MYC over expression in tumor tissues is well-established by many studies ⁶. We calculated the ratio (tumor / native) of MYC expression (FPKM) in our dataset. We found that in 63 pairs of patient tumor tissues (87.5%), MYC expression is more than 2 times higher than that in the corresponding native tissues (Response Fig. 12).

Response Fig. 12 Dot plot to show the fold change of MYC gene expression between paired tumor and native tissues from each patient. One dot indicates the fold change of the paired tissues from one patient .

Fig.1F: The authors mention that they identify 11796 new enhancers in their 74 CRC samples. They don't mention that they missed 18477 enhancer (uniquely found by Cohen et al in 20 cell lines) and that the overlap (24K) between the two datasets is smaller than the sum of their unique enhancers (30K). A more informative analysis would again be showing a correlation matrix between all samples and the published COAD cell lines ChIPseq results.

Response: The purpose of this analysis is to indicate that the enhancer profile of CRC patients is partially different from the profile of CRC cell lines, which could highlight the meaning of this study. The numbers here represent that of significant enhancers, which emerged at least in two tissues. If we count the number for all identified enhancers, it will be much larger.

According to your advice, we merged all the significant enhancer regions of our tumor samples and the 20 cell lines we used here, and then generated a correlation matrix between these two sets of data (Response Fig. 13). The result indicates that the tissue data has relatively low correlation with cell line data, but they have relatively high correlation within themselves.

Response Fig. 13 Heatmap for the spearman correlation of enhancer profiles among all CRC patient tumor samples and 20 CRC cell lines.

In Fig 2 1100 lost and 5590 gained VELs are introduced, determined as discussed in the text and in Fig. S3. Earlier they show a different analysis (Fig. 1F), mentioning 11K lost and 19K gained enhancers/. What is the value of Fig.1F (of using different methods and mentioning different numbers of differential H3k27Ac sites)?

Response: We utilized our H3K27ac data from patient tumor and native tissues to identify VELs, and the purpose of this analysis was to find out the enhancer alteration between these two kinds of tissues. The analysis in Fig. 1F was performed with two different sets of CRC data, from patient tissues and cell line respectively, and the purpose of the analysis has mentioned in the last question. The detailed information of VEL identification could be found in the Method/Identification of VELs section.

Please explain in the text how VELs are coupled to genes (linear distance? First neighboring gene?).

Response: We utilized the closest gene, i.e. the first neighboring gene as the associated gene for a specific VEL. And for the identification of VSEL-associated genes, we have described in the section of Method /Identification of genes associated with VSELS.

Fig.S3F: “the magnitude of the change in expression positively correlated with the number of VELs per gene (Extended Data Fig. S3F).the number of associated gained VELs”. Staring at Fig. S3F I wonder: is this significant? The authors may consider checking whether this correlation is better appreciable when considering the (delta) ChIPseq signal strength (ChIP counts) rather than

the number of VELs?

Response: We performed the corresponding analysis as you mentioned, and the result is shown below (Response Fig. 14). The result indicates that the values of Δ VEL and Δ associated gene expression are not strictly correlated, but the overall trend is still positively correlated ($R = 0.37$). That may have two reasons:

- 1) Histone acetylation on enhancers usually represents the accessibility of enhancer DNA sequences, which is related with its potential bound by other proteins. Although it is often considered to be related with transcription factor activity, actually it is regulated by many factors, sometimes, even by transcription repressors.
- 2) As it is difficult to identify VEL associated genes accurately, we used the closest genes of VELs as their associated genes. It is common when 3D genome data are not available. Then the fact is that sometimes some genes and enhancers are mis-matched, which may cause bias for the study.

Response Fig. 14 Scatter plot for the correlation between H3K27ac signal alteration (RPM) and differential gene expression (FPKM) of associated genes for all significant VELs. The black line represents regression line.

Fig.3B See before: please provide such spearman correlation matrix for all samples (CRC and control), based on all enhancers (not just VELs). Do we see the same clusters?

Response: The corresponding correlation matrix mentioned in your question could be found in Response Fig. 10. We also generated a correlation matrix (based on all significant tumor enhancers) for tumor samples by the CMS classification ordered in Fig. 3B. The result indicated that no matter using VELs or all enhancers, the enhancer profile correlation matrix followed the similar trend, i.e. CMS2 was the most homogenous subgroup, and CMS4 was the most heterogenous one (Response Fig. 15).

Response Fig. 15 Spearman correlation matrix of all tumor significant enhancer signal among all CMS classification identified patients, samples are divided by specific CMS subgroups.

Fig.3E: it is not clear to me whether all VELs were taken here, irrespective of whether they are found in the given subgroup? If so, this does not much over Fig. 3D?

Response: Yes, for Fig. 3E (Fig. 3F in the revised manuscript), we considered all the gained VELs we identified for all four subgroups, so the reference enhancers are the same for every subgroup. And Fig. 3D (Fig. 3E in the revised manuscript) provided different information from Fig. 3E (Fig. 3F in the revised manuscript), which showed the specific significant gained VEL number of four subgroups.

The authors use gene Ontology and a pathway analysis to analyze the genes linked to the VELs for the different subgroups. They conclude for example that subgroup 2 genes control WNT signaling and lipid metabolism. While this may be true, they have their RNA-seq data to verify this. Are the associated genes indeed specifically upregulated in the corresponding tumors? Is this true also for other VELs and their associated genes in the other MCS groups?

Response: For the genes involved in WNT signaling and lipid metabolism, we analyzed their expression in both tumor and native samples of CMS2. The result showed that a large portion of genes related to these two biological functions are up-regulated in tumor samples (Response Fig. 16).

The reason we just showed the functional annotation of target genes of CMS2-specific gain VELs was that the numbers of same VELs in other 3 subgroups were too small to analyze (Fig. 3G in revised manuscript). About the genes related with Wnt signaling and lipid metabolism, we compared their expression between all 4 different subgroups (Response Fig. 17). We can see that a large portion of them (more than 50%) expressed highest in CMS2 subgroup.

Response Fig. 16 Scatter plot showing the expression of genes related with lipid metabolism and Wnt signaling in both tumor and native tissues.

Response Fig. 17 Heatmaps to show the relative expression level of genes related with lipid metabolism and Wnt signaling.

The authors then repeat the exercise, now focusing on super enhancers (VSELs). This leads to the identification of three (not four) subgroups. Which classification (VEL or VSEL based) should we find most meaningful? Is there any correlation between the four subgroups scored by VELs versus the three found by VSELs?

Response: We think the reviewer might have some misunderstanding about our result. Here, we just tested the possibility to do classification with VSELs, but we are still far away from a good classification method. It requires much more work to achieve a successful classification. The R package CMScaller (Eide, Peter W et al. "CMScaller: a R package for consensus molecular

subtyping of colorectal cancer pre-clinical models.” Scientific reports vol. 7,1 16618. 30 Nov. 2017, doi:10.1038/s41598-017-16747-x) was used in this process, which is designed for analysis based on gene expression profiles but not with VEL. More detailed information could be found in Method “CMS classification for tumor tissues” section.

More generally: the authors promote the importance of considering enhancer profiles to classify tumors. Enhancers control gene expression, of which RNAseq is the direct read-out. When clustering on RNAseq (on differentially expressed genes), are the same four sub-groups identified? If not, what is the value of classifying on enhancers? If yes, what is the added value of classifying on enhancers?

Response: As described previously, CMS classification (using the R package CMScaller) is based on gene expression profiles of all samples, but not enhancers. We did some attempt on classifying by VSEL, and this part of results showed in Extended Data Fig. S8D. CMS classification is the most popular classifying method for CRC, and its criterion is based on the phenotype of microsatellite instability (MSI) and the mutation of some key genes (e.g. APC, KRAS and TP53)⁷. And later the R package was developed to do classification with gene expression profiles.

Enhancers control gene transcription, but enhancer activity is not simply equal to mRNA level. Transcription of one single gene could be regulated by multiple enhancers; and in different cell types, when the mRNA level is similar, the corresponding active enhancer could be different^{6,8,9}. Usually, enhancer profiles are much more complicated than gene expression profiles. That is the reason why enhancers are considered to be useful in cell identity. The purpose of the current analysis is to identify enhancer traits for known CRC sub-groups, and putative novel cancer-related genes, molecular targets and pathways.

For the value of the results in Extended Data Fig. S8D, we think it showed the potential to provide do classification for CRC tumors based on several key super enhancers. But we found it did not work ideally with the current effort, because it divided all tumors into three subgroups and the traits among different subgroups were not very obvious, which did not show advantage compared with current classification methods. A good classification method should be able to identify novel subgroups and useful to clinical studies. So, we just showed the data in supplementary figures. But it is still possible to improve the current methods with selected enhancer subsets in the future.

The authors perform CRISPRi to silence super enhancers in the HCT116 cell line and show that surrounding genes are downregulated. Is the opposite also true: do the associated genes in the tumor samples show specific upregulation in the corresponding tumors? This seems more relevant information, as we want to know the function of these SVELs (and VELs) in vivo.

Response: We selected the genes tested in our dCas9-KRAB perturbation assay to show their gene expression in the tumor and native tissues of the current study (Response Fig. 18). All the genes are up-regulated in tumor tissues.

Response Fig. 18 Box plot for gene expression (FPKM) of 10 VSEL associated genes (SLC7A5, RNF43, CYP2S1, CEBPB, LIF, IER3, VEGFA, TBC1D16, TNFRSF6B and PHF19) in both tumor and native tissues.

The authors probably refer to the wrong figure panel (4E should be 4F?), when stating “The difference of proliferation was not very significant (Data not shown), however, quite a few cell lines exhibited attenuated migration ability, including PHF19, LIF, SLC7A5, CYP2S1, RNF43, VEGFA and TBC1D16 (Fig. 4E).”

Response: Thank you for pointing out our mistake. We have corrected it.

In the abstract it says: “Further experiments verified the functions of 6 super enhancers governing PHF19, LIF, SLC7A5, CYP2S1, RNF43 and TBC1D16 in regulating cancer cell migration.” I am not an expert on migration assays, but I wonder whether the assay performed on a single cell line HCT116, and the results shown in Fig. 4F, are sufficient to draw this conclusion.

Response: We have performed additional experiments during revision in multiple cell lines and xenograft models (Fig. 4G-I, Extended Data Fig. S9 & S11). The results still supported our conclusions.

The 3C data are not essential for the paper, in my mind, but are also not of sufficient quality and, as presented, don’t allow drawing conclusions on looping. 3C is an very difficult to control assay: each given ligation product is extremely rare and consequently, accurate quantification of such rare products is very difficult. The control used (fragment-internal primerpair) correct for differences in amounts of template, but not for differences in crosslinking, digestion and ligation. More importantly, there is no correction carried out for differences in amplification efficiencies between primer pairs, which strongly impact results. Having a control template with equal amounts of all analyzed ligation products, to determine for each pair its amplification efficiency and correct for this, is crucial for correct interpretation. Finally, results need to be plotted on top of the loci, such that the chromosomal distances between analyzed fragments become appreciable. In B (CEBPB), provided that results after normalization will look the same (which I predict is unlikely)it is unclear why P3 and P4 (both on the enhancer) are so different in contact. In C (VEGFA) it currently is strange to see such big differences between P3 and P4 (that are extremely close on the chromosome). In D (CYP2S1) the most distal fragment P1, which seems upstream of the enhancer, appears to loop to the gene: why. In C and D, to appreciate a loop, you would want to analyze two fragments in between the gene and enhancer.

Response: We agree with the reviewer that 3C is an assay difficult to control. The result is affected by multiple factors, such as cross-linking, enzyme digestion efficiency, site selection, chromatin structure and primer efficiency. The primer pair with highest signal is not definitely matched to the enhancer site. Our data here were compared with a primer pair on *GAPDH* gene without DpnII cut sites, which is commonly used in many 3C studies and represents the background signal of the experiment. Our results here demonstrated that the identified enhancers are close to their target gene promoters spatially, which is important to support the connection between them. We have added the primer information in the revised manuscript to clarify it.

Fig.5B (and C): “Heatmap of transcription factors ranked by predicted core regulatory circuitry (CRC) total degrees (Tumor - Native tissue). Top 30 tumor and native specific TFs were listed.” Please explain in the text what this analysis entails.

Response: The analyzing details could be find in Method “Core regulatory circuitry for super enhancer associated TFs” section, the definition of IN and OUT degree for TF were also described there. The method was published by Dr. Ricky Young before ¹⁰. Briefly, for each SE-regulated TF, these criteria were quantified by measuring the inward binding of other SE associated TFs (IN degree) and the outward binding of the TF to other SEs (OUT degree). This regulatory circuitry across all SE-associated TFs in CRC identified cliques of TFs with similar patterns of IN/OUT degree, strong interconnectivity, and higher likelihoods of motif co-occurrence at enhancers. We picked out top 30 tumor and native specific TFs from these cliques as the candidates for further screening.

It states: “the gene expression analysis based on TCGA datasets suggested ASCL2 was highly expressed specific in colorectal cancer (Extended Data Fig. S11E).” Why based on TCGA? The authors generated their own RNAseq data and should discuss how ASCL2, and the other identified factors shown in Fig. 5C, express in their own control and CRC tissues. Are they upregulated in their CRCs? If not, it is difficult to understand how they would suddenly act as drivers.

Response: Here we demonstrated that ASCL2 is a CRC specific TF and showed its higher expression in our patient tissues in Extended Data Fig. S11E (Extended Data Fig. S12E of the revised manuscript), at the left two boxes. To further support it, we utilized TCGA public gene expression profile data to verify our conclusion (boxes at right). In Fig. 5C, we have also shown the gene expression related information. The y axis in this scatter plot represents the gene expression fold change (tumor / native) of the specific TFs listed in Fig. 5B, and the circle size indicates the mean expression (FPKM) of TFs in its specific tissue (red for tumor and blue for native tissue).

The added value of mapping enhancers in tumors, as compared to a transcriptome analysis, to me seems not to come from an improved ability to classify tumors (discussed above), but indeed to learn from the VELs and VSELs the underlying motifs, which can help identifying driver transcription factors. Only Figure 5 is dedicated to this exercise. The top hit, ASCL2 (Fig.5C), is mentioned but ignored in follow up studies and instead four TFs are selected (RUNX1, MAZ,

MAFK and KLF3) for a limited analysis (proliferation and migration assay upon knockdown in one cell line). These selected factors, curiously, don't show up from their own ChIP-seq analysis (Fig.5C), but were all (except for RUNX1) selected from literature. This twist is hard to understand. If the authors want to make the point (as they do) that systematic enhancer mapping is important to understand the biology of tumors, they should seriously characterize their top hit TFs and show that indeed these are key factors in tumorigenesis.

Response: As we responded above, enhancer information actually provides a more complicated layer of transcription information other than transcriptome. Our study is the first time to identify active enhancers in a large cohort of CRC tissues. Although, our sample number is still not big enough and the classification method is not well developed. But our analysis has shown that enhancer information has the potential to contribute to CRC classification. Besides the above and TF prediction, VELs and VSELs are useful to determine the functions of specific non-coding elements in cancer cells, which are often found mutated in tumor tissues.

About TF investigation, the reason we did not further study ASCL2 is that many reports have well established its role in CRC. In comparison, the oncogenic role of KLF3 has not been reported yet and only very few reports have studied RUNX1 and MAZ. For TF prediction, we used two different approaches (Fig. 5A-C). All the four TFs emerged in our analysis. During revision, we have performed additional experiments to further demonstrate KLF3 as an oncogenic TF in CRC (Fig. 5D-H, Extended Data Fig. S14).

The abstract claims: "we identified KLF3 as a novel oncogenic transcription factor in CRC." I don't think the limited data (Fig.5D-F) on a cell line allow drawing this conclusion. Also, it is unclear where KLF3 suddenly comes from: it does not show up among the top candidates (Fig.5C), and the KLF motif is hardly enriched (Fig. 5A).

Response: During revision, we have performed additional experiments to study the role of KLF3 in CRC. All the results are consistent with our previous conclusion (Fig. 5D-H, Extended Data Fig. S14). For motif analysis, the p values should not be used to represent the importance of TFs, since the analysis is largely dependent on the existing ChIP-Seq data. Well-characterized TFs usually show better results in prediction, and ChIP-Seq data for most TFs are still very limited.

Reviewer #4 (Remarks to the Author): Expert in CRC genomics and whole-genome sequencing

Li et al present an impressive amount of data which could really improve our current understanding of the mechanisms underlying CRC. However, I feel the current way of analysing the data could be improved and the RNAseq data could be used more efficiently.

Response: We appreciate the reviewer for his positive comment to our work.

Specific comments:

1) ChIP-seq analysis:

- Why were paired tumor and normal ChIP-seq data not used together in the MACS2 pipeline? Normal samples could be used as the control sample which would automatically yield a list of differential peaks between tumor and normal for every patient. From this list the authors could extract VELs and VSELs per patient and determine their recurrence in the total cohort. To me this seems a more elegant and statistically sound approach than the one the authors currently use, in which arbitrary and debatable cut offs for fold changes are used to define VELs and VSELs.

Response: Thanks for your advice, but this approach is not suitable for our situation. As a general rule of thumb, MACS2 is a useful tool for peak calling, but not designed to handle the direct comparison of two ChIP samples. It only works when one assumes there are no peaks identified in the sample acting as control but not for the sample acting as treatment. MACS2 does have its own differential peak calling tool, but it has been shown to only work with narrow peaks and tends to be very stringent. The case in which we can successfully compare two ChIP samples in MACS2 is when we had a histone modification like H3K4me3 as a treatment and H3 as the control.

- The authors indicate multiple times that patients were excluded from certain analyses because low numbers of VELs or VSELs were found. However, when the data generated passes the quality control (so sufficient numbers of unique reads, good phred scores etc) they cannot be excluded from further analysis for the simple reason that their results (from the comparison tumor vs normal) are not what the authors were expecting. The reasoning that this may represent a sampling problem may be true but cannot be made conclusive when no HE slides have been evaluated for the tissues.

Response: We have explained why we deleted some samples during certain analysis in the Method (section “ChIP-seq data processing” and “Identification of VELs”), but we are willing to provide more detailed explanation here to clarify the issue.

The 9 deleted samples consist of two parts. 4 samples (patient #20, #21, #22, #24) were deleted because of their low number of significant H3K27ac peaks (Extended Data Fig. S3A). Based on our experience and some published papers, we set 2500 as the number threshold, and we could see that the peak number of these 4 patient native tissues is lower than the line. If the significant peak number of one ChIP-seq data is much lower than the expected number, it's likely that the ChIP quality is not good enough. We thought this might be a sampling problem, because these 4 samples are native tissues, and they often have much less cell number than tumor tissues, a nightmare for ChIP assay with tissues. We have checked our H3K27ac ChIP-seq experiments for these 4 samples, including the concentration of the ChIP DNA, Phred score and the sequencing reads number. Q30 for these four samples are normal; #20, #21 and #22 reads number are relatively low but still OK for analysis (< 20M); and the ChIP DNA concentration of all four samples are much lower than others. So, we decided to delete the 4 samples for further analysis. If we keep them for VEL identifying, we'll get lots of false candidates. However, their paired tumor data are normal and can be used in other analysis.

The reason for ruling out the other 5 patients (patient #8, #19, #38, #41, #57) in VEL identifying is that there is little difference between the paired tissues of individual VEL number. We mentioned in the manuscript that the first step we calling VEL is to identify all VELs for every patient, the cut-off is Fold Change > 2. For individual patient, if their VEL number (GAIN VEL + LOST VEL) is less than 500, we treated them as the “No difference samples” (Extended Data Fig. S3B). The purpose of VEL identification is to find the difference between tumor and native tissues, so these “No difference samples” were almost useless in the analysis. If we keep them in the following processes, they may influence the result of statistical test (calculating padj and recurrence). Based on these reasons, we decided to exclude the 5 patients in VEL identifying process.

- CMS specific VELs should be identified by a statistical comparison between the H3K27Ac signal between CMS groups, instead of only looking at a fold change.

Response: We have reconsidered the p-value of H3K27ac for CMS specific VEL candidates between subgroups, and the cut-off of p-value is 0.05. Based on this added criterion, we revised the Fig. 3G in our revised manuscript (Response Fig. 19).

Response Fig. 19 Subgroup specific GAIN VEL number for 4 CMS subgroups.

2) RNAseq analysis:

- The authors state that 20M reads were generated for every sample, however I assume that also here there will be a range (similar to the 30-50M reads for ChIP-seq). Please indicate this in the relevant section.

Response: We have shown this information of all samples in Supplementary Table 2 “Information of all sequencing data”. You can find the corresponding reads number of all ChIP-seq and RNA-seq data there.

- DEGs were identified using DESeq2. Was a paired t-test used for this? The authors write they used contrast adjustment for multiple groups comparison. I think this indicates that they used the multifactor setting to control for the patient effect, which effectively comes down to a paired analysis. Could the authors please clarify this?

Response: Thanks for your question. We indeed didn't describe the process of DEGs identification

very clearly. Actually, we considered the pairs information of all sample when using DESeq2, and used a design of the form “~ patient + condition” to account for the pairs when providing the Sample Table. The “patient” column was used to record the patient identifier and the “condition” column to indicated the tissue type. So paired t-test was used to identify DEGs.

- CMS calling: the authors need to indicate in the manuscript how many CMS1, CMS2, CMS3, CMS4 and no group samples were identified in their cohort (this is now only given in the supplementary figures). This is important for readers to judge whether CMS2 is really the only group showing specific VELs or whether this is due to sample numbers (since CMS2 is the largest group). In addition, I feel that the “no group” samples should still be included in the analysis and it seems they are currently left out?

Response: Thanks for your advice. We have moved Extended Data Fig. S4A to Fig. 3A in our revised manuscript, and added the identifier of no group samples. For the “no group” samples, we performed corresponding analysis as Extended Data Fig. S5B, and we found heterogeneity existed among the samples based on their enhancer H3K27ac signal (Response Fig. 20). Considering that CMScaller may exist some bias in identifying CMS subgroup samples (based on gene expression), we didn't include “no group” samples into our previous analysis.

Response Fig. 20 Heatmap showing the correlation of H3K27ac signal at specific gain VELs in tumor samples of “no group” subgroup. Correlations were calculated by Spearman correlation coefficient.

- As far as I can tell the RNAseq data was not used to verify whether the identified VELs and VSELs actually have an effect on the expression of their associated SE-genes. This analysis would really add to the manuscript as it would demonstrate the effect of the differentially activated V(S)ELs in CRC.

Response: Actually, for VELs, we have performed corresponding analysis to clarify the relationship between VELs and the expression of their associated genes (Extended Data Fig. S3F). As for the VSELs, we have supplemented this analysis and added it to our revised manuscript (Response Fig. 20, Extended Data Fig. S8CG). Here we identified the proximal genes as the SE-associated genes. We can see that the mean expression of most VSEL-associated genes corresponded to their traits, i.e. gain-VSEL-associated genes were up-regulated and lost-VSEL-

associated genes were down-regulated in tumor tissues. To evaluate potential bias in the analysis, we also counted their percentage of significant DEGs ($\log_2\text{FC} > 1$ and $p.\text{adj} < 0.01$). According to the result, 66 gain-VSEL-associated genes (21.1%) were higher expressed DEGs in tumor, and 45 lost-VSEL-associated genes (37.5%) were higher expressed DEGs in normal tissues.

Response Fig. 20 Scatter plot showing the mean gene expression of gain and lost VSEL-associated genes in tumor and native tissues.

3) General remarks:

- In the cluster analysis identifying 3 CRC subgroups it would be interesting if the CMS group would also indicated per sample to see whether the subgroups based in V(S)ELs are overlapping with the CMS subgroups or not.

Response: Thanks for your advice. We have integrated the two classifying information and the result was shown below (Response Fig. 21). Most of the CMS2 samples are overlapped with G1 subgroup based on VSELs, but others had no such trend. We have added the of result into our revised manuscript as Extended Data Fig. S8F.

Response Fig. 21 Bar plot showing the overlap between our cluster analysis identified samples and CMS subgroup samples.

- Loss of *KLF3* was previously shown to be associated with more aggressive CRC (PMID: 28423541). This seems to be somewhat contradictory to the current findings. Could the authors comment on that?

Response: We have analyzed the expression of *KLF3* in our data and TCGA data. We did not observe a significant difference between the adjacent and cancer tissues in our collection; among TCGA samples, *KLF3* expression was higher in tumor tissues but not significant enough. We then

compared the expression between the adjacent and cancer tissues of four CMS groups. Interestingly, *KLF3* is lower expressed in the cancer tissues of CMS1 group, and higher expressed in CMS3 group (Response Fig. 22). Although our sample number is a little bit small when split into four groups, the results suggest that *KLF3* may play different roles in the four groups. It will be interesting to determine *KLF3*'s function in each CRC subgroup.

Response Fig. 22 Expression of *KLF3* in four CMS subgroups.

Reference

- Jacobs, R. J., Voorneveld, P. W., Kodach, L. L. & Hardwick, J. C. Cholesterol metabolism and colorectal cancers. *Curr Opin Pharmacol* **12**, 690–5 (2012).
- Ye, J., Liu, S., Shang, Y., Chen, H. & Wang, R. R-spondin1/Wnt-enhanced *Ascl2* autoregulation controls the self-renewal of colorectal cancer progenitor cells. *Cell Cycle* **17**, 1014–1025 (2018).
- Jubb, A. M. *et al.* Achaete-scute like 2 (*ascl2*) is a target of Wnt signalling and is upregulated in intestinal neoplasia. *Oncogene* **25**, 3445–57 (2006).
- Wei, X. *et al.* *Ascl2* activation by YAP1/KLF5 ensures the self-renewability of colon cancer progenitor cells. *Oncotarget* **8**, 109301–109318 (2017).
- Wang, X. *et al.* RNA sequencing analysis reveals protective role of kruppel-like factor 3 in colorectal cancer. *Oncotarget* **8**, 21984–21993 (2017).
- Lancho, O. & Herranz, D. The MYC Enhancer-ome: Long-Range Transcriptional Regulation of MYC in Cancer. *Trends Cancer* **4**, 810–822 (2018).
- Dienstmann, R. *et al.* Consensus molecular subtypes and the evolution of precision medicine in colorectal cancer. *Nat Rev Cancer* **17**, 268 (2017).
- Villar, D. *et al.* Enhancer evolution across 20 mammalian species. *Cell* **160**, 554–66 (2015).
- Shlyueva, D., Stampfel, G. & Stark, A. Transcriptional enhancers: from properties to genome-wide predictions. *Nat Rev Genet* **15**, 272–86 (2014).
- Saint-Andre, V. *et al.* Models of human core transcriptional regulatory circuitries. *Genome Res* **26**, 385–96 (2016).

REVIEWERS' COMMENTS

Reviewer #1 (Remarks to the Author):

My concerns have been addressed.

Reviewer #2 (Remarks to the Author):

The authors did a thorough job addressing my previous concerns to the best of their capability. I am supportive of publication of this paper, which will provide a nice dataset to the CRC research community.

Reviewer #3 (Remarks to the Author):

I thank the authors for their extensive rebuttal and I appreciate their efforts to address my comments. There are a few remaining items that I find need to be resolved, including some explanatory text and graphs which the authors provided to the reviewers, but that should be incorporated in the actual manuscript.

1. Response Fig.9: please transfer both panels to Fig. S2. I know this information is also in the suppl. Tables, but these two panels enable the reader to immediately grasp the variation/quality of the data.
2. Same for Response Fig 10 and Response Fig 11: please show side-by-side in a supplemental figure (eg Fig. S2).
3. Please add your response "We found that in 63 pairs of patient tumor tissues (87.5%), MYC expression is more than 2 times higher than that in the corresponding native tissues" to the main text.
4. Please add and briefly discuss response Fig. 13 to the supplemental data (I agree with the authors that this supports the value of their efforts to characterize primary tumor tissue instead of cell lines).
5. Please add response Fig.17 to the supplementals, and briefly discuss
6. 5. Please add response Fig.18 to the supplementals, and discuss
7. I pointed out: The 3C data are not essential for the paper, in my mind, but are also not of sufficient quality and, as presented, don't allow drawing conclusions on looping. 3C is an very difficult to control assay: each given ligation product is extremely rare and consequently, accurate quantification of such rare products is very difficult. The control used (fragment-internal primerpair) correct for differences in amounts of template, but not for differences in crosslinking, digestion and ligation. More importantly, there is no correction carried out for differences in amplification efficiencies between primer pairs, which strongly impact results. Having a control template with equal amounts of all analyzed ligation products, to determine for each pair its amplification efficiency and correct for this, is crucial for correct interpretation. Finally, results need to be plotted on top of the loci, such that the chromosomal distances between analyzed fragments become appreciable. In B (CEBPB), provided that results after normalization will look the same (which I predict is unlikely) it is unclear why P3 and P4 (both on the enhancer) are so different in contact. In C (VEGFA) it currently is strange to see such big differences between P3 and P4 (that are extremely close on the chromosome). In D (CYP2S1) the most distal fragment P1, which seems upstream of the enhancer, appears to loop to the gene: why. In C and D, to appreciate a loop, you would want to analyze two fragments in between the gene and enhancer.
Somewhat surprisingly, the authors did not take this comment very seriously. They state that they

agree controlling 3C is difficult, but (a) they don't incorporate the controls that I explained are necessary to conclude anything from 3C, and (b) they don't change the plotting of results necessary to properly interpret the results. Without these two crucial controls/items, their conclusion "Our results here demonstrated that the identified enhancers are close to their target gene promoters spatially" is incorrect.

Authors, either perform carefully controlled 3C experiments (as explained above and in authoritative methods papers), or leave out the 3C data and the claims about looping/spatial proximity.

Reviewer #4 (Remarks to the Author):

I would like to thank the authors for their detailed responses to my raised comments. I am quite satisfied with their answers and only have 2 minor suggestions:

1) I would recommend to include the used test (paired t-test) in the description in the materials & methods section regarding the identification of the DEGs.

2) With respect to KLF3, I find it quite interesting that its role may differ depending on the CMS subtype. I understand numbers are small, but still it may be worthwhile to mention this in the discussion I think. Additionally, as Pubmed shows me their already is one publication on KLF3 in CRC I wonder why this is not mentioned in the manuscript.

** See Nature Research's author and referees' website at www.nature.com/authors for information about policies, services and author benefits

To whom it may concern,

Thank you for giving us a chance to submit our revised manuscript, titled “*Genome-wide profiling of active enhancers in colorectal cancer*” (NCOMMS-20-47734A). We have made all the corrections as suggested. Below is our point-by-point response to each comment, in which the text in blue is the original comments from the reviewers and our response in black.

Reviewer #3 (Remarks to the Author):

I thank the authors for their extensive rebuttal and I appreciate their efforts to address my comments. There are a few remaining items that I find need to be resolved, including some explanatory text and graphs which the authors provided to the reviewers, but that should be incorporated in the actual manuscript.

1. Response Fig.9: please transfer both panels to Fig. S2. I know this information is also in the suppl. Tables, but these two panels enable the reader to immediately grasp the variation/quality of the data.

Response: We have added Response Fig. 9 into the manuscript as Extended Data Fig. S2H.

2. Same for Response Fig 10 and Response Fig 11: please show side-by-side in a supplemental figure (eg Fig. S2).

Response: We have added Response Fig. 10 & 11 into the manuscript as Extended Data Fig. S2I&J.

3. Please add your response “We found that in 63 pairs of patient tumor tissues (87.5%), MYC expression is more than 2 times higher than that in the corresponding native tissues” to the main text.

Response: We have added the sentence at line 151 as suggested.

4. Please add and briefly discuss response Fig. 13 to the supplemental data (I agree with the authors that this supports the value of their efforts to characterize primary tumor tissue instead of cell lines).

Response: We have added Response Fig. 13 into the manuscript as Extended Data Fig. S2K.

5. Please add response Fig.17 to the supplementals, and briefly discuss

Response: We have added Response Fig. 17 into the manuscript as Extended Data Fig.

S6M.

6. 5. Please add response Fig.18 to the supplementals, and discuss

Response: We have added Response Fig. 18 into the manuscript as Extended Data Fig. S10B.

7. I pointed out: The 3C data are not essential for the paper, in my mind, but are also not of sufficient quality and, as presented, don't allow drawing conclusions on looping. 3C is an very difficult to control assay: each given ligation product is extremely rare and consequently, accurate quantification of such rare products is very difficult. The control used (fragment-internal primerpair) correct for differences in amounts of template, but not for differences in crosslinking, digestion and ligation. More importantly, there is no correction carried out for differences in amplification efficiencies between primer pairs, which strongly impact results. Having a control template with equal amounts of all analyzed ligation products, to determine for each pair its amplification efficiency and correct for this, is crucial for correct interpretation. Finally, results need to be plotted on top of the loci, such that the chromosomal distances between analyzed fragments become appreciable. In B (CEBPB), provided that results after normalization will look the same (which I predict is unlikely)it is unclear why P3 and P4 (both on the enhancer) are so different in contact. In C (VEGFA) it currently is strange to see such big differences between P3 and P4 (that are extremely close on the chromosome). In D (CYP2S1) the most distal fragment P1, which seems upstream of the enhancer, appears to loop to the gene: why. In C and D, to appreciate a loop, you would want to analyze two fragments in between the gene and enhancer.

Somewhat surprisingly, the authors did not take this comment very seriously. They state that they agree controlling 3C is difficult, but (a) they don't incorporate the controls that I explained are necessary to conclude anything from 3C, and (b) they don't change the plotting of results necessary to properly interpret the results. Without these two crucial controls/items, their conclusion "Our results here demonstrated that the identified enhancers are close to their target gene promoters spatially" is incorrect. Authors, either perform carefully controlled 3C experiments (as explained above and in authoritative methods papers), or leave out the 3C data and the claims about looping/spatial proximity.

Response: We understood the reviewer's concern and have removed the 3C data from the manuscript.

Reviewer #4 (Remarks to the Author):

I would like to thank the authors for their detailed responses to my raised comments. I am quite satisfied with their answers and only have 2 minor suggestions:

1) I would recommend to include the used test (paired t-test) in the description in the materials & methods section regarding the identification of the DEGs.

Response: We have added the information to the Method/ RNA-seq data processing and DEG identification section.

2) With respect to KLF3, I find it quite interesting that its role may differ depending on the CMS subtype. I understand numbers are small, but still it may be worthwhile to mention this in the discussion I think. Additionally, as Pubmed shows me their already is one publication on KLF3 in CRC I wonder why this is not mentioned in the manuscript.

We have added the reference and discussed the possibility in the discussion section.

Editorial Requests

Response: About the requests in the Policies and Checklists, we have responded one by one in the tables. A revised summary report was also prepared according to the editor's comments. The checklist and summary report were uploaded as separate files. About the requests to the specific text and figures, we have made changes as below, For Request 1-4, we have either prepared new panels as requested, and added the requested description into the corresponding figure legends.

For Request 5, we have added one section titled "Statistics and Reproducibility" in the methods section.

For request 6, we have removed the restriction to our GEO datasets, and added the GEO number of other public datasets in the section of "Data Availability".

For Request 7, we have added on supplementary figure (Sup. Fig. S15) to present our original films of the requested WB data.

For request 8, scale bars have been added into the mentioned panels.